# Learning the fitness dynamics of pathogens from phylogenies

Noémie Lefrancq[1,2,3✉], Loréna Duret[1], Valérie Bouchez[4,5], Sylvain Brisse[4,5], Julian Parkhill[2,6] & Henrik Salje[1,6]

The dynamics of the genetic diversity of pathogens, including the emergence of lineages with increased fitness, is a foundational concept of disease ecology with key public-health implications. However, the identification of such lineages and estimation of associated fitness remain challenging, and is rarely done outside densely sampled systems[1,2]. Here we present phylowave, a scalable approach that summarizes changes in population composition in phylogenetic trees, enabling the automatic detection of lineages based on shared fitness and evolutionary relationships. We use our approach on a broad set of viruses and bacteria (SARS-CoV-2, influenza A subtype H3N2, *Bordetella pertussis* and *Mycobacterium tuberculosis*), which include both well-studied and understudied threats to human health. We show that phylowave recovers the main known circulating lineages for each pathogen and that it can detect specific amino acid changes linked to fitness changes. Furthermore, phylowave identifies previously undetected lineages with increased fitness, including three co-circulating *B. pertussis* lineages. Inference using phylowave is robust to uneven and limited observations. This widely applicable approach provides an avenue to monitor evolution in real time to support public-health action and explore fundamental drivers of pathogen fitness.

Most pathogens have constantly changing strain compositions. Pressures to evade host immunity, environmental shifts or changing abilities to infect and disseminate in hosts result in the emergence of some lineages with increased fitness and the extinction of others. These dynamic patterns of genetic diversity are a fundamental aspect of disease ecology. They also have potentially critical public-health implications, because such changes could signify, for example, escape from vaccine-derived or innate immunity or improved transmissibility. It has, however, been difficult to identify lineages with differential levels of fitness, that is, different abilities to spread in the population, especially outside highly genetically sampled pathogen systems such as SARS-CoV-2 or influenza[1–3]. Identifying lineages with improved fitness at the population level would enable a focused public-health response, through, for example, targeted vaccination and could provide key insights into the underlying ecology of disease systems.

Existing methods to monitor the fitness of strains at the population level rely on independently defined strain or clade definitions, for example, Pango lineages[4] or NextStrain clades[5], the global clades for influenza[6], or strains defined by predetermined single mutations for *B. pertussis*[7]. Strain fitness can be estimated using models that capture the changing proportion of individual lineages through time, typically with multinomial logistic models. These models are computationally efficient and provide key insights, for example, into the effect of amino acid substitutions[8] or vaccine implementation[3,9] on fitness. These approaches rely on the ability of researchers to group individual

sequences into different lineages. However, such groupings are usually based on consensus opinion and arbitrary thresholds in amino acid differences and, importantly, are unlinked to underlying differences in fitness. This is problematic because it means we are not reliably capturing emergent lineages with increased fitness.

Phylogenetic-tree-based methods provide an alternative strategy for uncovering strain fitness. Strains with increased fitness will transmit more frequently, leading to a higher branching rate in the phylogeny and more sampled descendants. The fitness of lineages can therefore be inferred from their branching pattern in a phylogeny using phylodynamic approaches such as birth–death models[10]. Multi-type birth–death models extend this idea by allowing the birth and death rates of lineages, and thereby their fitness, to depend on the state or type of the lineage, which may be known (such as genotype, mutations[11,12]) or inferred[13]. However, these models are computationally challenging to run, especially given the large amount of data now being generated. They are also susceptible to sampling biases in both space and time, which are common in phylogenetic analyses. There are alternative approaches that focus on the broad population structure[14] or changes in effective population size[15] but models are not able to capture lineage fitness. Other studies[1,10,16] have been done at a more granular level, but these do not allow for a broad understanding of fitness changes through time.

Here we present phylowave, an agnostic approach that summarizes the changes in population composition in phylogenetic trees through

[1]Department of Genetics, University of Cambridge, Cambridge, UK. [2]Department of Veterinary Medicine, University of Cambridge, Cambridge, UK. [3]Department of Biosystems Science and Engineering, ETH Zürich, Basel, Switzerland. [4]Biodiversity and Epidemiology of Bacterial Pathogens, Institut Pasteur, Université de Paris, Paris, France. [5]National Reference Center for Whooping Cough and Other Bordetella Infections, Paris, France. [6]These authors jointly supervised this work: Julian Parkhill, Henrik Salje. ✉e-mail: ncmjl2@cam.ac.uk

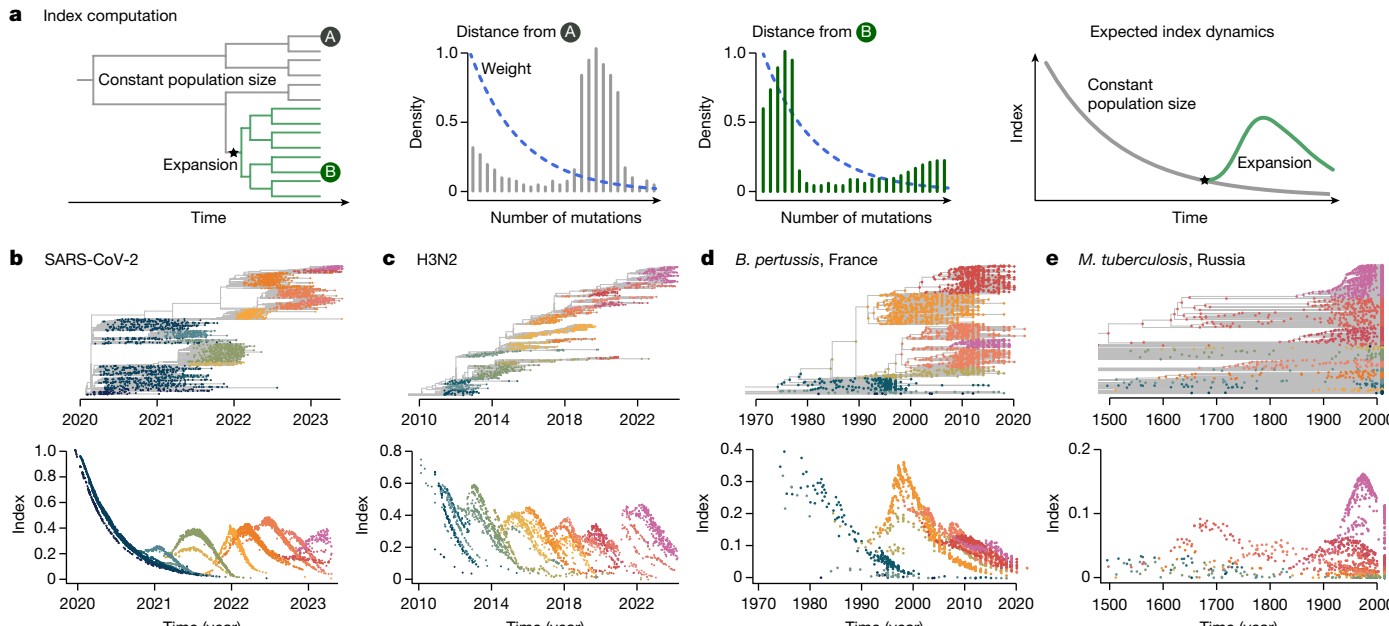

**Fig. 1 | Tracking changes in population composition by following index dynamics. a**, Schematics describing the principles of index computation. From left to right: example of a time-resolved phylogenetic tree with a background population (grey) and an emerging lineage (green); pairwise distance distribution from terminal node A, or terminal node B, respectively, to the rest of the population, with the dashed blue line denoting the geometric weighting; and expected index dynamics over time (see Methods for details). **b**–**e**, For each pathogen, SARS-CoV-2 (**b**), influenza A/H3N2 (H3N2; **c**), *B. pertussis* (**d**) and *M. tuberculosis* (**e**) we present the index dynamics computed at each node (terminal or internal). Colours represent the different lineages identified by their different index dynamics (Extended Data Fig. 3). Dynamics coloured by known lineages are presented in Extended Data Fig. 2.

time, enabling the automatic detection of circulating lineages based on fitness differences, which we quantify and link back to specific amino acid changes. We initially explore the robustness of our approach using a simulation study in which the underlying fitness difference between strains is known. We then apply this approach to SARS-CoV-2, influenza A subtype H3N2, *B. pertussis* and *M. tuberculosis*. We selected these respiratory pathogens as they present a diverse set of viruses and bacteria at both local and global scales, and include both well-studied and understudied threats to human health. Taking each pathogen in turn, we use phylowave to gain critical insights into the set of discrete lineages circulating over time, their individual fitness, as well as the genomic changes linked to quantified shifts in fitness.

## Agnostic identification of lineages

Phylowave builds on a genetic-distance-based index that measures the epidemic success of each node (internal or terminal) in a time-resolved phylogeny[16] (Fig. 1a). This measure is based on the expectation that nodes sampled from an emerging fitter lineage will be phylogenetically closer than the rest of the population at that time, as they will all share the same recent ancestor. The index of each node is derived from the distance distribution from that node to all other nodes that circulate at that time, weighted by a kernel with a set timescale. This weight allows us to track lineage emergence dynamically, focusing on short distances between nodes (containing information about recent population dynamics) rather than long distances (containing information about past evolution). The timescale is tailored to the specific pathogen studied and its choice will depend on the molecular signal, as well as on the transmission rate. Here we used timescales ranging from months (typical of RNA viruses) to years (typical of bacteria). Using the principles of coalescent theory in structured populations[17–19], we derive the expected index dynamics through time in the case of an emerging successful lineage (Fig. 1a, derivation in Supplementary Information section 1). Once we have calculated the index value for each sequence, we implement a tree-partitioning algorithm using generalized additive

models that finds the set of lineages (that is, groups of tips and nodes) that best explains the observed index dynamics.

To quantify the fitness of each lineage, we developed a multinomial logistic model to fit the proportion of tips and nodes that belong to each lineage through time. We assumed each lineage has a constant fitness through time, defined as its relative growth rate in the population. By taking into account lineage emergence based on the most recent common ancestor (MRCA) of the lineages, our model does not estimate proportions for lineages that do not exist yet in the population, as opposed to implementations in other studies[8].

To assess the performance of our approach, we repeatedly simulated phylogenetic trees in which one lineage expands with a known fitness advantage compared with a background population (Extended Data Fig. 1). We found that phylowave was indeed able to identify fitter emerging lineages (Extended Data Fig. 2a–c). Its ability to recover lineages depends on the time between emergence and the dates of sequences, with lineages with only small fitness advantages requiring sequences that cover longer time periods (Extended Data Fig. 2e). Nevertheless, in cases in which the fitness difference between the two strains was greater than 0.02 per year, our method was able to consistently identify the emerging lineage, noting that for lineages with lower fitness advantages the time to become the dominant lineage is greater than 200 years. We further found that the sampling intensity was substantially less important than the sampling period (Extended Data Fig. 2f). Finally, we compared phylowave with alternative approaches (fastbaps[14] and treestructure[15]) (Supplementary Fig. 1). Although treestructure and fastbaps found the emerging lineage in some cases, they always found additional lineages in the phylogenies that did not have any true selective advantage (Supplementary Fig. 2).

## Application to pathogens

We applied phylowave to four viral and bacterial pathogens: SARS-CoV-2 (*n* = 3,129 global whole-genome sequences), H3N2 (*n* = 1,476 global haemagglutinin (HA) sequences), *B. pertussis* (*n* = 1,248 whole-genome

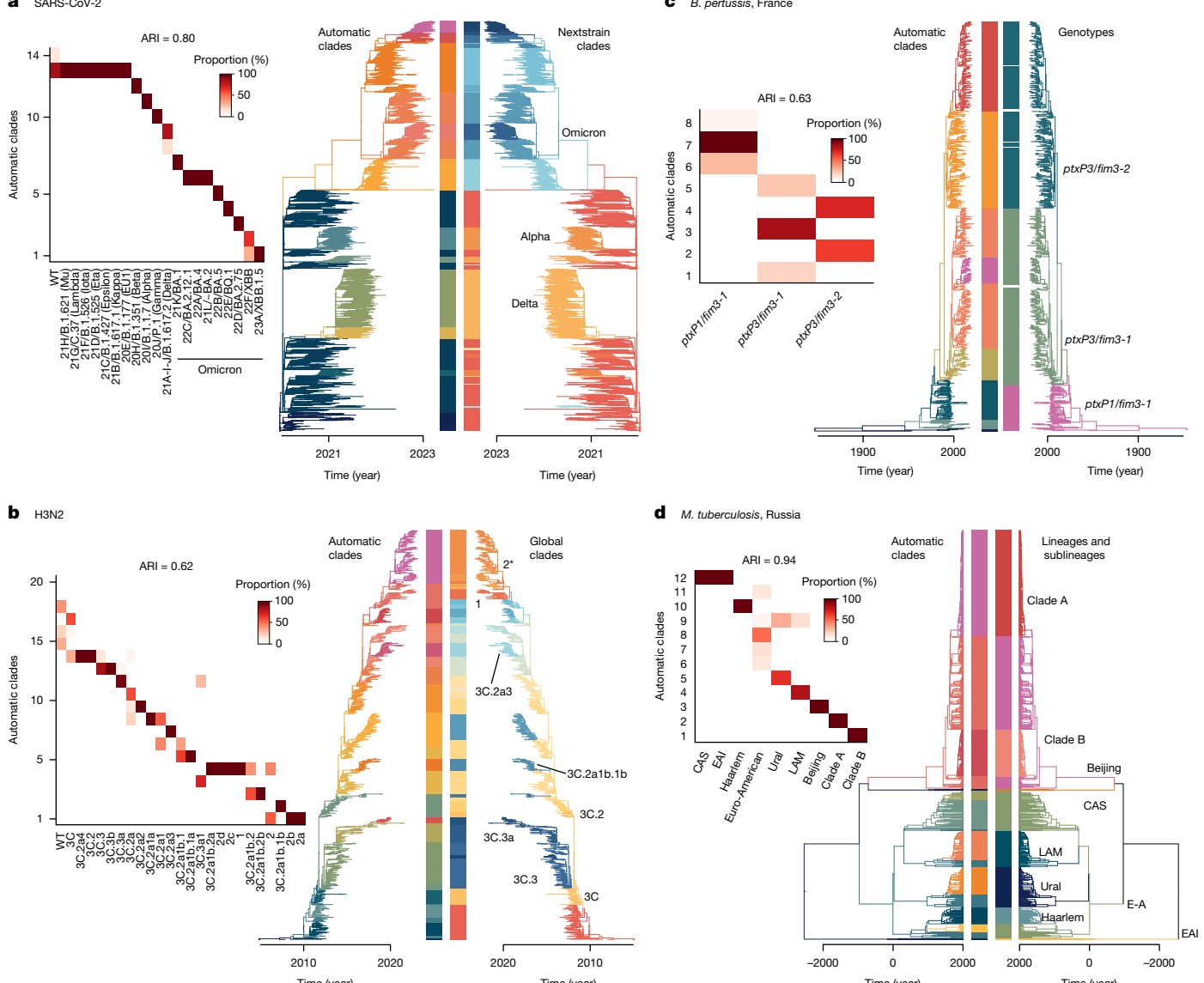

**Fig. 2 | Comparison of the identified lineages to the known population composition. a–d**, present time-resolved phylogenetic trees and heat maps for SARS-CoV-2 (**a**), H3N2 (**b**), *B. pertussis* (**c**) and *M. tuberculosis* (**d**) to compare the identified lineages with the automatic clades found by phylowave. Darker colours in the heat maps represent more agreement between both classifications. Contingency tables are presented in Supplementary Tables 2–5.

The heat maps are presented in large in Extended Data Fig. 5. Time-resolved phylogenetic trees are coloured by respective lineage classifications: automatic clades on the left and previously identified lineages on the right. The colours of the automatic clades are the same as in Fig. 1. CAS, Central Asian strain lineage; EA, Euro-American lineage; EAI, East African Indian lineage; LAM, Latin American–Mediterranean lineage.

sequences from France) and *M. tuberculosis* (n = 998 whole-genome sequences from Samara, Russia[20]) (Fig. 1b–e and Extended Data Figs. 3 and 4). We found that for each pathogen considered, phylowave produced evidence of lineages with clear fitness differences, as evidenced by subpopulations of genetically related strains with discrete index dynamics (Fig. 1b–e). Taking each pathogen in turn, we compared our lineage assignments with existing lineage definitions.

We computed the adjusted rand index (ARI) to measure the agreement between classifications, accounting for random clustering[21] (Fig. 2 and Extended Data Fig. 5). An ARI value of 1 corresponds to perfect agreement with previously defined lineages, whereas a value of 0 would be expected if clusters were assigned at random. We found that across the pathogens the level of concordance was high (ARI range 0.62–0.94). For example, the previously defined SARS-CoV-2 variants of concern (Alpha (B.1.1.7; 20I), Beta (B.1.351; 20H), Gamma (P.1.*; 20J), Delta (B.1.617.2/AY.*; 21A/21J) and Omicron (BA.1.1.529/BA.*; 21K)) and other

previously defined subvariants closely matched phylowave-defined lineages[22,23] (Fig. 2a and Extended Data Fig. 5). Lineages generated by fastbaps[14] (v.1.0.8) and treestructure[15] were less consistent with these predefined lineages (Supplementary Fig. 3). The existing definitions of global H3N2 clades also closely matched phylowave lineages (for example, 3C.3a, 3C.2a3 and 3C.2a1b.1b), with the occasional discrepancy in the exact node of emergence (e.g. 3C, 3C.2 and 3C.3). Phylowave also identified previously defined *B. pertussis* clades (ARI = 0.63), including those defined by changes in alleles of the promoter of the pertussis toxin (*ptxP*) and fimbriae 3 gene (*fim3*)[7]. Furthermore, phylowave identified three extra *B. pertussis* lineages with clear distinct index dynamics (Fig. 1d, pink, red and purple lineages) that have not been previously identified. Finally, we recovered the known lineages and sublineages (ARI = 0.92) that were present in the *M. tuberculosis* dataset[20,24,25].

Across the pathogens, previously defined subvariants that reached a maximum prevalence of less than 5% at any time in the datasets

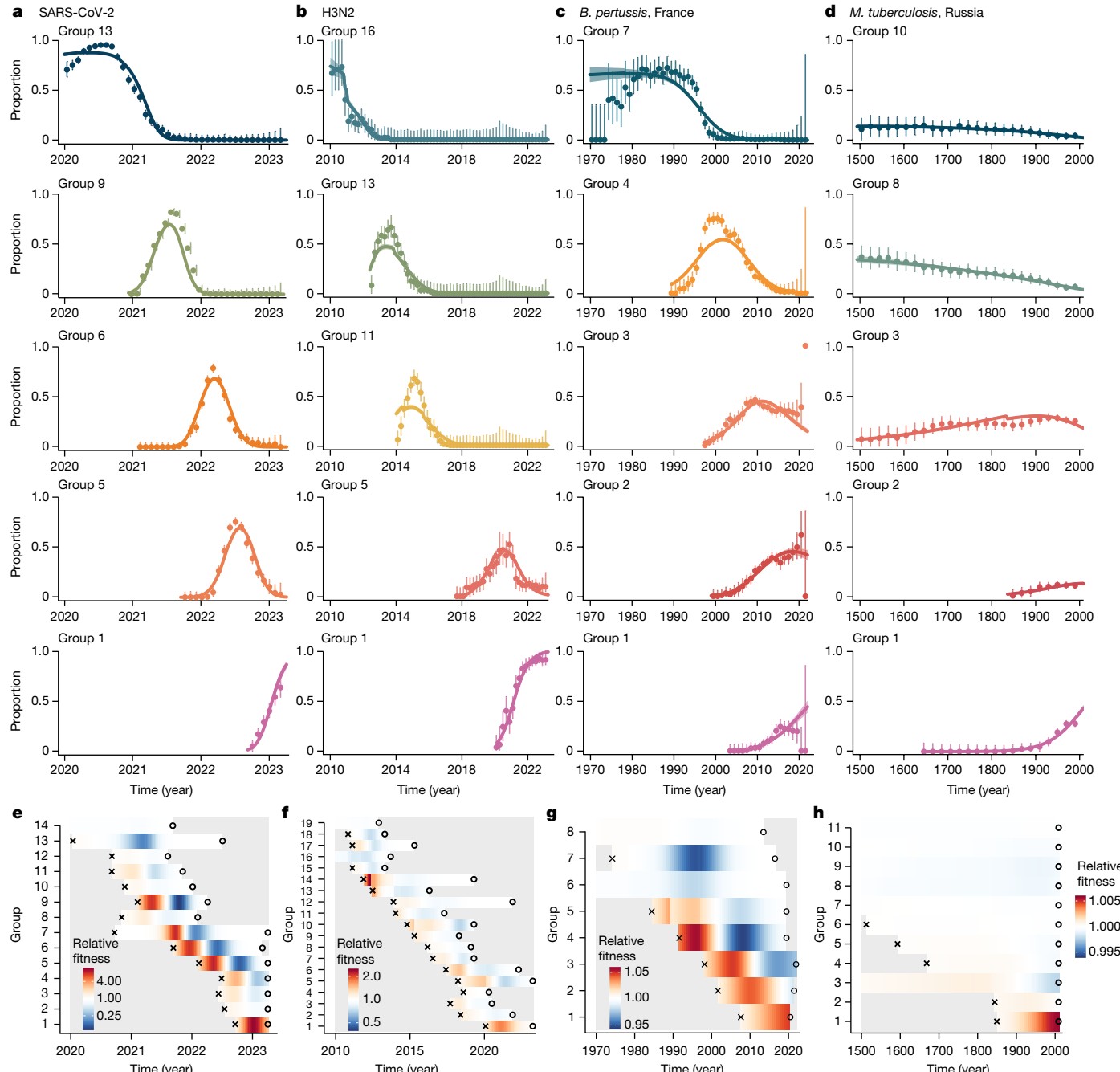

**Fig. 3 | Estimation of the fitness of each lineage. a–d**, Model fits per pathogen: SARS-CoV-2 (**a**), H3N2 (**b**), *B. pertussis* (**c**) and *M. tuberculosis* (**d**). For each pathogen, we present the fits for the five most prevalent groups. The fits for all groups are presented in Supplementary Figs. 4–7. Coloured dots represent data, bars denote 95% confidence intervals. Coloured lines and shaded areas represent the median and 95% credible interval of the posterior. **e–h**, Relative fitness of SARS-CoV-2 (**e**), H3N2 (**f**), *B. pertussis* (**g**) and *M. tuberculosis* (**h**) over time. Estimates for all groups are presented in Extended Data Fig. 7. Crosses indicate the MRCA of each group. Open circles indicate the last isolate from each group in our datasets.

were generally not identified by phylowave (for example, Eta/B.1.525, Mu/B.1.621 and EU1 for SARS-CoV-2, clades 1* of H3N2 and Central Asian Strain and East African Indian *M. tuberculosis* lineages). The exact limits when phylowave can identify discrete lineages will depend on the underlying prevalence, the level of sampling and fitness differences. For example, replicating the SARS-CoV-2 analysis by continent, we obtained phylowave lineages that matched previously identified variants of interest that were mainly contained to those continents and that we did not identify when using the global dataset (such as Eta/B.1.525 in Africa, Mu/B.1.621 in the Americas and EU1 in Europe)[26–28] (Extended Data Fig. 6). These findings show that previous attempts to identify

discrete lineages often resulted in lineage classifications with distinct fitnesses, even if the various algorithms used to define the lineages did not include fitness as a metric.

We next estimated the fitness of each lineage using our logistic growth model. This simple model was able to capture the lineage dynamics of each pathogen, despite substantially different trends across the pathogens investigated (Fig. 3a–d and Supplementary Figs. 4–7). We found that the underlying fitness of each emerging lineage was non-null, in line with the lineages identified having true different levels of fitness (Extended Data Fig. 7). We further computed the inferred real-time fitness of each lineage in the population. Although

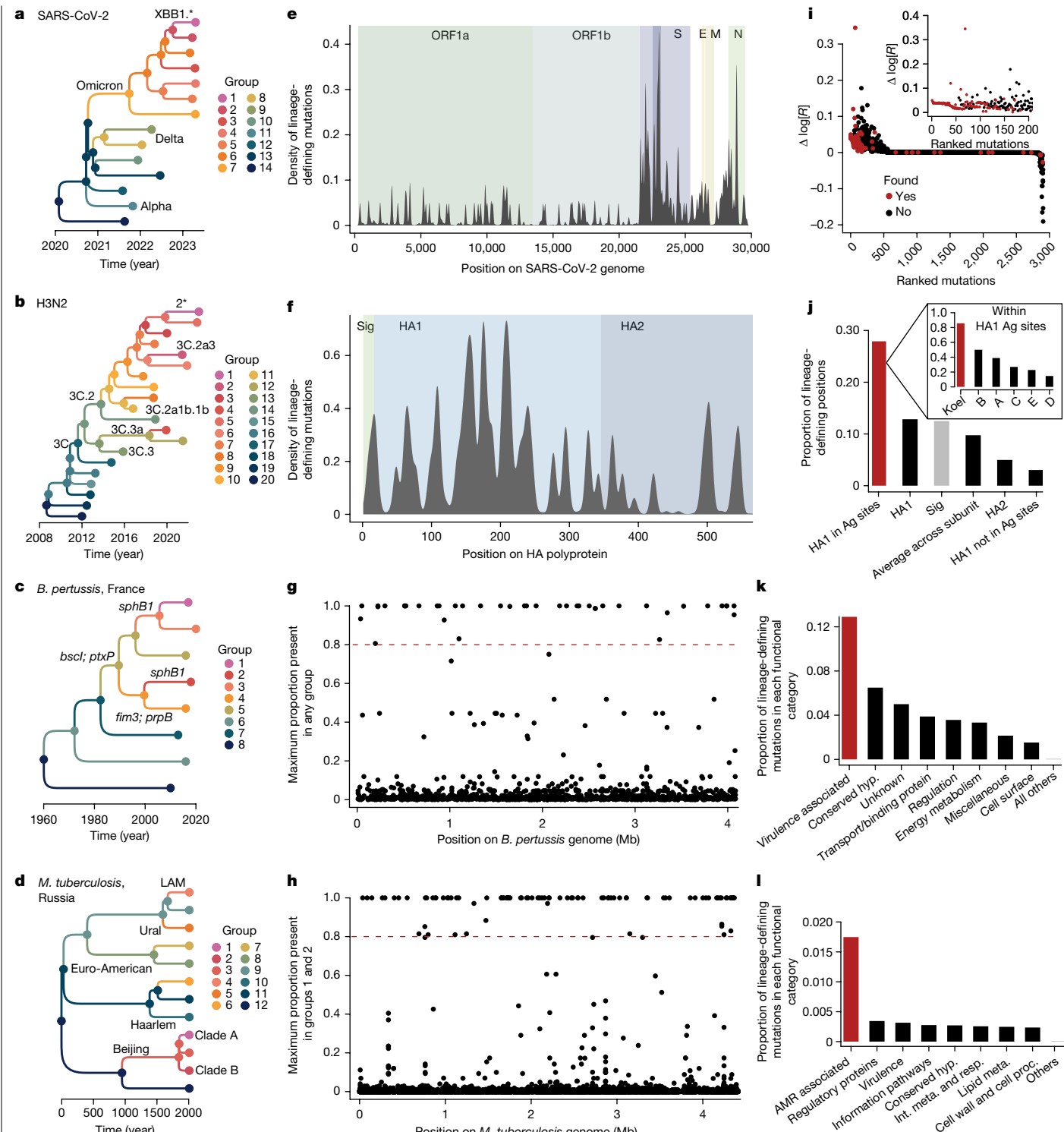

**Fig. 4 | Lineage-defining genetic mutations.** For each pathogen, we present a summary of the genetic evolution of the lineages. **a–d,** For each pathogen, we present the lineage trees representing the genealogical relationship between them. Key clades are highlighted for SARS-CoV-2 (**a**), H3N2 (**b**), *B. pertussis* (**c**) and *M. tuberculosis* (**d**). Colours indicate groups. **e–h,** Lineage-defining mutations along the genome of each pathogen considered. **e,f,** For SARS-CoV-2 (**e**) and H3N2 (**f**) viruses, we plot the density of lineage-defining mutations along the full genome (SARS-CoV-2) or HA polyprotein (H3N2). For SARS-CoV-2, colours indicate the main open reading frames (ORFs): S, spike; E, envelope; M, membrane; and N, nucleocapsid. For H3N2, colours indicate the HA subdomains. Sig, signal peptide. **g,h,** For *B. pertussis* (**g**) and *M. tuberculosis* (**h**) we plot for each mutation the maximum proportion of that mutation that is present in any group (*B. pertussis*) or in groups 1 and 2 (*M. tuberculosis*). The dashed lines represent the 0.8 cut-off. **i–l,** Functional relevance of the mutations identified. **i,** For SARS-CoV-2, we compare the substitutions analysed previously[8] and the substitutions found to be lineage defining in our study. For each amino acid change, the figure presents the estimated increased in fitness $\Delta \log R$ (*y* axis), as a function of its rank, inferred by statistical significance (*x* axis). Mutations in red are found by our method. **j,** For H3N2, we plot the proportion of positions that are lineage defining in each HA polyprotein subunit and antigenic sites[32,33]. Ag site, antigenic site. **k,** For *B. pertussis*, we plot the proportion of mutations that are lineage-defining in each functional category[35]. Conserved hyp., conserved hypothetical. **l,** We plot the proportion of mutations that are lineage defining in each functional category for *M. tuberculosis*[36]. Int. meta. and resp., intermediary metabolism and respiration; proc, processes. The lists of lineage-defining mutations for each pathogen can be found in Supplementary Tables 6–9.

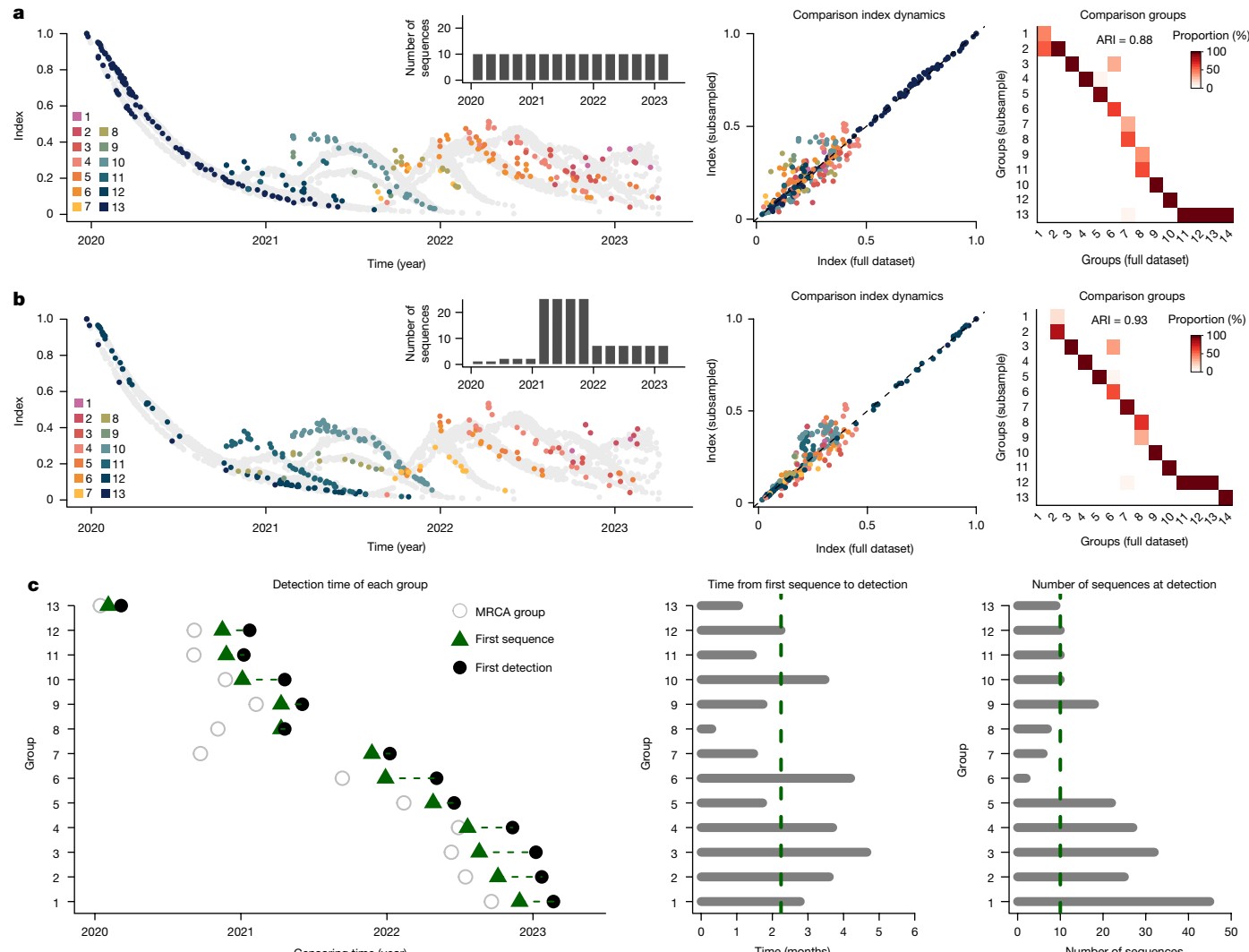

**Fig. 5 | Robustness of phylowave to sampling intensities and time to lineage detection. a,b**, Robustness to downsampling. We kept only 150 sequences from the global SARS-CoV-2 tree, either sampled uniformly through time (**a**) or in a temporally uneven manner (**b**). From left to right: index dynamics computed on the subsampled trees, coloured according to detected lineages, and the temporal distribution of the sequences (insets); pairwise comparison of the index computed at nodes (internal and terminal) in the trees from the full dataset (*x* axis) and subsampled datasets (*y* axis); heat map comparing the automatic clades found by phylowave for the full dataset (*x* axis) to the automatic clades found for the subsampled datasets (*y* axis). Darker colours on the heat map denote more agreement between both classifications. **c**, Time to lineage detection. The full global SARS-CoV-2 dataset was censored every two weeks and we reran the detection algorithm. From left to right: detection time of each group, with the open circles denoting the MRCA of each group in our tree, the green triangles denoting the first sequence of the group in our dataset and the black dots denoting the first detection of the group by phylowave; time from first sequence isolated in our dataset to group detection; number of sequences in each group at the time of detection. The dashed lines denote the median time to detection or the number of sequences at detection.

our model estimates a constant fitness parameter for each lineage, their actual fitness through time depends on which other lineages are circulating at that time. For SARS-CoV-2, we found that lineage 1, corresponding to Omicron XBB1.5, had the best maximal real-time fitness, followed by lineages 5 and 7, corresponding to Omicron BA.5 and BA.1 (Fig. 3e and Extended Data Fig. 7). The fitness of H3N2 lineages was more homogeneous across the population, with lineages persisting for 3.9 years on average after their emergence[29,30] (Fig. 3f and Extended Data Fig. 7). For *B. pertussis*, our results are consistent with those of previous studies[3], noting that three lineages (labelled 1, 2 and 3) emerged after the implementation of a new acellular vaccine in France[31] in 1998 (Fig. 3g and Extended Data Fig. 7). These three lineages had the highest fitness of all *B. pertussis* strains, pointing towards immune pressure on lineage dynamics from the new vaccine. *Mycobacterium tuberculosis* lineage fitness was the most stable of the four pathogens explored, reflecting its long-established diverse population. The only

exception is the comparatively recent emergence[20] of lineages 1 and 2 (Fig. 3h and Extended Data Fig. 7). These lineages are rising sharply in the population and have a relative fitness per year of 1.0057 (95% credible interval 1.0055–1.0060) and 1.00087 (95% credible interval 1.00077–1.00098) for lineage 1 and 2.

## Lineage-defining mutations

We next explored whether specific changes in the genomes were linked to lineage fitness by identifying lineage-defining mutations (Fig. 4). We defined such mutations as: (i) present in at least 80% of the sequences in that lineage and (ii) not present in the ancestral lineage. We did not infer lineage-defining mutations directly from the tree because there might be some uncertainty in the exact timing of the emerging lineage, and the tree itself will depend on sequencing intensity over time. In particular, a lineage can take some time to start growing and become detectable.

We instead used a comparison of sequences between different lineages to identify the specific mutations that are different. Although we focus on mutations, we note that phylowave can be applied to other covariates, for the analysis of both genotypes (for example, insertions or deletions and gains or losses of genes) and phenotypes (such as resistance to antimicrobial drugs). For each pathogen, we looked at where those mutations are located in their genomes, and how functionally relevant each of them is. For SARS-CoV-2, we found that the highest density of lineage-defining amino acid substitutions was located in the receptor binding domain of the spike protein, with low densities in ORF1a and ORF1b and none in ORF10 (Fig. 4a,e,i and Supplementary Figs. 8 and 9). Our lineage-defining mutations were consistent with those described in a previous analysis that estimated nucleotide positions linked with shifts in fitness for six million SARS-CoV-2 genomes[8]. We found that our screening recovered all of the 55 fittest mutations and 86% of the top 100 fittest mutations (Fig. 4i). The mutations missed by our method are mainly linked to small subclades of variants and they seem to have spread in those clades only. We obtained similar results for H3N2, for which most of the lineage-defining amino acid substitutions are located in the HA1 domain (Fig. 4b,f,j and Supplementary Fig. 10). We then investigated specifically whether the substitutions that we found were located in previously described antigenic sites[32]. We found that the antigenic sites had the highest proportion of amino acid substitutions compared with the rest of the gene, and that of those, the Koel sites had the highest proportions of substitutions[33] (Fig. 4j). Among the Koel sites, 86% of positions (*n* = 6, out of 7) were recovered by phylowave. The only position missed, 155, is the oldest variable position, and is not covered by our dataset. We also recovered the main previously described *B. pertussis* lineage-defining mutations, namely in *ptxP* and *fim3* (Fig. 4c,g,k). Furthermore, we found a selection of other associated mutations that had not been described previously, with two distinct non-synonymous mutations in *sphB1* being of particular interest as they suggest convergent evolution (Supplementary Fig. 11). *sphB1* encodes a protease that is involved in the extracellular release of the pertussis filamentous haemagglutinin, a *B. pertussis* acellular vaccine antigen and key host interaction factor[34]. Overall, we found that virulence-associated genes had the highest proportion of lineage-defining mutations (Fig. 4k). We also investigated the mutations associated with the most recent clades of *M. tuberculosis* (clades 1 and 2 in Fig. 3h). As reported previously[20], we found that genes associated with antimicrobial resistance had the highest proportion of lineage-defining mutations (Fig. 4d,h,l and Supplementary Fig. 12).

### Tracking lineages in real time

Phylowave enables us to track changes in the population composition over time, with a direct link to fitness. As our method relies on the estimation of the pairwise distance distribution of each node in a tree, the number of sequences does not affect the index dynamics, as long as sequences are representative of the diversity (Fig. 5a). To demonstrate this robustness to sampling biases over time, we conducted a sensitivity analysis using the SARS-CoV-2 dataset by repeatedly removing a subset of genomes, including in a temporally uneven manner. We then re-estimated the circulating lineages each time. We were still able to detect nearly all the lineages, even when using heavily biased datasets (Fig. 5b, mean ARI = 0.90). As with other phylodynamic methods, we note that phylowave is sensitive to biases in the sequence source in cases in which the sequenced pathogens are not representative of the diversity of the underlying population. Finally, we explored how quickly phylowave was able to detect newly emerging lineages. We truncated our full global SARS-CoV-2 dataset every two weeks and reran the detection algorithm. We found that our model was able to capture each lineage, with a median delay of 2.2 months after emergence, with only ten sequences required (Fig. 5c). Considering that the SARS-CoV-2 dataset used in this study comes from NextStrain and was composed of only 3,129 sequences (approximately 0.02% of all sequences available on the Global Initiative on Sharing All Influenza Data (GISAID) at the time of the study), the time to lineage identification could be further shortened with larger datasets.

## Conclusion

In this study, we presented an approach that can agnostically track changes in population composition in phylogenetic trees, even in situations in which the availability of sequences is heavily biased. Across a broad range of pathogens, we have shown that we can recover the main known circulating lineages for each pathogen, as well as identify previously unknown lineages that have substantial changes in fitness. We can quantify the relative fitness of each lineage and identify genetic changes linked to the emergence of new, fitter lineages. This approach can have important implications for public-health surveillance. There is increased interest in the systematic sequencing of pathogens detected in healthcare settings. By integrating such sequencing efforts into phylowave, public-health agencies will be able to identify emergent strains in a timely manner and the data can be used to promote targeted interventions. Phylowave is also able to make fundamental insights into pathogen ecology. By quantifying the relative fitness advantage of new strains, phylowave can help us to identify potential drivers of emergence, including the role of population immunity derived from natural infection or vaccination. Finally, by identifying the specific genomic changes linked to fitness changes, this work provides testable biological hypotheses about genetic variants in each pathogen that are driving the changes in population fitness of that pathogen.

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

## Methods

### Sequence data

For each pathogen, we compiled a dataset to investigate the changes in the population composition. For SARS-CoV-2 and H3N2, we extracted the datasets from the publicly available NextStrain[37] time-resolved phylogenies, accessed on 14 April 2023. These datasets are sub-samples from all publicly available sequences in GISAID, to represent the diversity as much as possible (we used the 'all-time' dataset for SARS-CoV-2 and the '12 y' one for H3N2). In all, we have 3,129 whole-genome SARS-CoV-2 sequences sampled from 26 December 2019 to 3 April 2023, and 1,476 H3N2 HA sequences from 1 January 2005 to 3 April 2023 (Supplementary Tables 10 and 11). For B. pertussis, we used 1,248 sequences from 1953 to 2022, collected by the National Reference Center for Whooping Cough and Other Bordetella Infections in France (Supplementary Table 12). This dataset is composed of 1,023 previously published sequences[3,38–41] and 225 newly sequenced isolates. The new isolates have been sequenced with the same methods as previously described[3]. This dataset is representative of the B. pertussis diversity in France as the National Reference Center receives isolates from 42 sentinel hospitals throughout France. For M. tuberculosis, we used 997 previously published sequences, isolated in 2008–2010 in Samara, Russia[20]. This dataset is also representative of M. tuberculosis sequence diversity at that location as isolates were prospectively collected from individual patients living in the region and representative of the entire population (Supplementary Table 13).

### Multi-sequence alignment for each pathogen

We compiled alignments of all sequences. For SARS-CoV-2, we used the precomputed multi-sequence alignment provided by GISAID. For H3N2, we aligned all HA sequences using MAFFT[42] (v.7.309) with default settings. We then manually checked that the alignment did not have any frameshift and had minimal gaps. For B. pertussis and M. tuberculosis, we worked from raw reads with a previously described protocol[3]. Reads were trimmed using Cutadapt[43] (v.3.4) and quality was checked with FastQC[44] (v0.11.9). Using BWA-MEM[45] (v0.7.17), reads were mapped against the complete Tohama I reference genome (accession number in RefSeq: NC_002929) or the complete H37Rv reference genome (accession number in RefSeq: NC_000962.3). Using GATK[46] (v.4.2.0.0), we kept variants that were present in at least 75% of reads, with a Phred quality score higher than 30, a minimum read depth of 5, a minimum mapping quality of 20 and a string odd ratio of less than 3. We masked all positions that were covered by less than five reads. Furthermore, we filtered out regions that are notoriously difficult to map and/or sequence, similarly to previous studies[3,47]. Namely, for B. pertussis we filtered out repeated regions (IS481, IS1002 and IS1663)[35] and phage regions using PHASTER[48]; for M. tuberculosis, we filtered out the functional categories 'PE/PPE' and 'insertion sequences and phages'[47]. For B. pertussis, we also checked for recombination in our alignment using Gubbins[49] (v.3.3.0). As a result, we obtained an alignment of 4,701 SNPs for B. pertussis and 30,533 SNPs for M. tuberculosis.

### Reconstruction of time-resolved phylogenies

For each pathogen, we obtained time-resolved phylogenies. For SARS-CoV-2 and H3N2, we used the NextStrain trees, accessed on 14 April 2023 (ref. 37). For B. pertussis and M. tuberculosis, we reconstructed the timed phylogenies specifically for this study using the SNP-based alignments. We first built maximum-likelihood trees using IQ-tree[50] (v.2.1.0) using a GTR + F + G substitution model. To assess the branch support, we used the ultrafast bootstrap approximation provided in IQ-tree, performing 1,000 replicates for each dataset with the bnni option to reduce the risk of overestimating the branch support[51].

For B. pertussis, the time tree was reconstructed using BEAST v.1.10.4[52], under a GTR substitution model[18] accounting for the number of constant sites, a relaxed lognormal clock model[53] and a skygrid population size model[54]. Three independent Markov chains were run for 150,000,000 generations each, with parameter values sampled every 10,000 generations. Runs were optimized using the GPU BEAGLE library[55] (v.4.0.0). Chains were manually checked for convergence (effective sample size values > 200) using the Tracer software[56] (v.1.7.2). We manually removed a 10% burn-in.

For M. tuberculosis, because all sequences were isolated in 2008–2010, we could not infer a clock rate, but instead, we used a previously estimated clock rate[57] of $4.6 \times 10^{-8}$ mutations per site per year. We used the software BactDating[58] (v.1.1) to perform a Bayesian reconstruction of the time tree. We used a fixed mean substitution rate, a relaxed clock rate and a constant effective population size. We ran the chain for 10,000,000 iterations and checked for convergence (effective sample size values > 200).

### Index definition

We developed an analytical approach that summarizes the changes in population composition in phylogenetic trees at every time point. Our approach builds on a genetic distance-based index, the timed haplotype density[16], that measures the epidemic success of individual sequences in a dataset. This measure is based on the expectation that sequences sampled from an emerging, fitter lineage will be phylogenetically closer than the rest of the population at that time, as they will all share the same recent ancestor. We extend this method to track population changes in phylogenetic trees through time.

We define the index of each isolate $i$ in its population at time $t$ as:

$$\text{Index}(i) = \sum_{d=0}^{\infty} D_i(d,t) b^d \tag{1}$$

Where $D_i(d,t)$ the distance distribution—in number of mutations or evolutionary time (branch length)—from isolate $i$ to the rest of the population (internal and terminal nodes) at time $t$ (Fig. 1) and $b^d$ is the kernel setting the weight of each distance $d$. $b$ is the bandwidth, $b \in [0,1]$, which is a parameter to set, linked to the timescale. We compute this index on each node in a tree (internal and terminal).

The weight allows us to track lineage emergence dynamically, focusing on short distances between nodes (containing information about recent population dynamics) rather than long distances (containing information about past evolution). The kernel is governed by the bandwidth $b$, which is a parameter to set. As $b$ is dimensionless, it is hard to set. Instead, we use the notion of timescale 50 to choose it: the time to the most recent common ancestor (TMRCA) was chosen such that pairs of isolates with shorter TMRCAs account for 50% of the kernel density[16]. This timescale is tailored to the specific pathogen studied and its choice will depend on the molecular signal, as well as the transmission rate. Here we used timescales ranging from 1.8 months (SARS-CoV-2, RNA virus) to 30 years (M. tuberculosis, a bacterium) (Supplementary Table 1).

Our approach provides a quantitative index value for each node (internal and terminal) in the tree, independently of any lineage classification providing an advantage over other methods that rely on lineage classification. This enables us to agnostically summarize the changes in population composition in phylogenetic trees at every time point.

Our definition is nearly the same as the one used in a previous study[16], with two critical differences: instead of computing the index by summing on each isolate in the population we now sum over the pairwise distance distribution, and we consider the collection time of each sequence to only compute the distance from $i$ to the rest of the population that is circulating at that time.

This index is similar to the local branching index[10], which is defined as the total surrounding tree length exponentially discounted with increasing distance from isolate $i$. In our case, rather than considering the tree length, we compute the distance between nodes.

This index definition enables us to write an expectation of the index dynamics over time, as theoretical pairwise distance distributions can be approximated for different populations. In practice, to compute the index of each node in a phylogenetic tree, we sum over the distance to all nodes in the population, rather than the distance distribution (see 'Index computation on time tree with sequences sampled through time' section).

## Linking the index dynamics to population history

The pairwise distance distribution $D_i(d,t)$ or more generally $D(d,t)$, can be seen as the probability, $P_c(s = \frac{d}{\mu l}, t)$, for any pair of sequences sampled at time $t$, to coalesce some time $s = \frac{d}{\mu l}$ in the past, with $\mu$ being the rate at which the pathogen accumulates mutations per site and per unit of time and $l$ the length of its genome.

$$D(d, t) = P_c\left(s = \frac{d}{\mu l}, t\right)$$

Therefore, at any time point, writing the probability of coalescing in the past enables us to compute the index in the population. We can update equation (1):

$$\text{Index}(t) = \int_0^{\mu l t} P_c\left(\frac{u}{\mu l}, t\right) \cdot b^u \, du \qquad (2)$$

We note that at time $t$, the maximum number of mutations accumulated is equal to $\mu l t$. For simplicity, we assume a linear accumulation of mutations through time in all the analytical expressions, although one could consider that mutations accumulate randomly given a Poisson distribution with rate $1/(\mu l t)$.

This probability $P_c(s = \frac{d}{\mu l}, t)$ is closely linked to the effective population size. In Supplementary Fig. 13, we show conceptually how, for different effective population sizes, the probability of coalescing changes, and how it affects the index dynamics. Formal derivations are presented in the Supplementary Information.

## Index computation on time tree with sequences sampled through time

We use equation (1) to compute the index of each node (internal or terminal) in a timed phylogenetic tree. To do this, for each node $i$, we compute its distance to all the other nodes present in the tree at that time (see Supplementary Fig. 14 for notations). All the nodes that fall in the time interval $[t_i - t_{\text{wind}}; t_i + t_{\text{wind}}]$ are considered to be circulating at the same time as $i$; with $t_i$ being the collection time of node $i$ and $t_{\text{wind}}$ the predefined time window width that is tailored to each pathogen. We also consider extant branches in the computation, as they are evidence of past circulation.

For computation efficiency, similarly to the previously published study[16], we then compute:

$$\text{Index}(i) = \sum_{j \in \text{nodes}} I(t_j > t_i - t_{\text{wind}} \text{ and } t_j < t_i + t_{\text{wind}}) d(i,j) b^{d(i,j)} \qquad (3)$$

Where nodes indicates the set of all nodes in the tree, and $I()$ is the indicator function.

This computation is efficient as it only requires the precomputation of the indicator function, the precomputation of the distance matrix and a matrix multiplication.

For the pathogens presented in our study we used the following parameters. SARS-CoV-2: a timescale of 0.15 years, and a window of time $t_{\text{wind}} = 15$ days. H3N2: a timescale of 0.4 years, and a window of time $t_{\text{wind}} = 0.25$ years. *B. pertussis*: a timescale of 2 years, and a window of time $t_{\text{wind}} = 1$ years. *M. tuberculosis*: a timescale of 30 years, and a window of time $t_{\text{wind}} = 15$ years.

We illustrate the impact of the timescale on the index dynamics in Extended Data Fig. 8 for the global SARS-CoV-2 tree.

To test the robustness of the index computation for the exact tree topology, we ran a sensitivity analysis on 3,000 trees sampled from the posterior of the BEAST run of *B. pertussis*. We chose *B. pertussis* for this analysis because it is the only pathogen for which we have a posterior distribution of trees. We repeatedly computed the index on each tree sampled from the posterior and computed the average index of each tip. Although there is uncertainty in the exact value of the index for each tip, we found that the index dynamics of each lineage remained very consistent across the posterior of trees (Supplementary Fig. 15).

## Agnostic detection of lineages

We develop an approach that is able to find the set of lineages in the tree that best explains the index dynamics. To do this, we build an algorithm based on generalized additive models (GAM) that jointly uses the phylogenetic relationships between nodes in the tree and their index.

In this section, for modelling purposes, we define lineages as monophyletic clades formed by one internal node and all its descendants. Here, these lineages can overlap, meaning that some isolates can be included in multiple lineages. We assume the tree to be binary. For a rooted binary tree with $n$ terminal nodes, there are $n - 2$ internal nodes that are not the root, and therefore $n - 2$ lineage possibilities—a substantial number. To keep the algorithm tractable, we limit the potential list of lineages to those starting with an internal node that has at least $N_{\text{off}}$ offspring, where $N_{\text{off}}$ is chosen. We note the set of internal nodes to test $\Pi$. Furthermore, to increase the accuracy of the detection, we take into account only internal nodes that have predefined characteristics. For *B. pertussis* and *M. tuberculosis*, as we constructed the bootstrap support of each node (see above), we consider only internal nodes that have a bootstrap support of at least 50% to be the potential start of lineages. This threshold is low, but effectively removes nodes that are not well supported. For SARS-CoV-2 and H3N2, instead of bootstrap support, we consider a minimum number of mutations. We consider only internal nodes that have a least one mutation on the branch directly upstream from them.

The algorithm models the log index through time. We use a log transformation to avoid having to restrict to model to positive values and to make sure the model does not overfit the index peaks.

The log index of each lineage $l$ is modelled using a cubic spline $S_l(t, k)$ with a predefined number of knots $k$. This allows us to model the log index of each node $i$, sampled at time $t_i$, given the lineage that it belongs to:

$$\log(\text{Index}_i) = \beta_0 + S_0(t_i, k) + \sum_{l=1}^{L} I(i \in l) S_l(t_i, k)$$

Where $\beta_0$ is the intercept, $L$ is the total number of lineages, $S_0(t, k)$ and $S_l(t, k)$ are penalized cubic regression splines with $k$ knots[59]. One null spline $S_0(t, k)$ is estimated to model the initial population, together with one spline for each of the $L$ lineages. If $L = 0$, then no $S_l(t, k)$ is estimated. $I()$ is the indicator function: $I(i \in l) = 1$ if $i$ is a member of the lineage $l$, and 0 otherwise.

In brief, the algorithm runs as follows. We start with a null model $M_0$ that fits the index dynamics with one spline $S_0(t, k)$ (that is, an unstructured population with one single index dynamic, $L = 0$). We store the deviance explained $\text{Dev}_0$ by model $M_0$. We then sequentially consider models with increasing complexity $M_L$: we start by first trying models with one lineage, $L = 1$. We go through the list of internal nodes $\pi$ that could be the start of a new lineage. When the deviance explained $\text{Dev}_1$ by the best model $M_1$ is higher than that of the previous null model $\text{Dev}_0$ we keep the lineage (effectively the node from $\pi$) that best explains the dynamics. We then continue this procedure for increasing $L$. For each number $L$, we go through the list of internal nodes $\pi$ that could be the start of a new lineage. When the deviance explained $\text{Dev}_L$ by the model $M_L$ is higher than that of the previous model $\text{Dev}_{L-1}$, we keep the lineage (effectively the node from $\pi$) that best explains the dynamics.

The algorithm is implemented in R v.4.1.2, using the package mgcv v.1.8 (ref. 60) to implement the GAM models.

As for any clustering algorithm, choosing the best number of lineages that describe the index dynamics is a challenging question. We took the approach of the elbow plot. We plot the deviances $Dev_L$ explained by each best model $M_L$, as a function of the number of lineages $L$. This approach enables us to see how well all the models are performing, and to choose the number $L$ of lineages at which the deviance explained does not increase substantially anymore (Extended Data Fig. 9). From this selected best number of lineages $L_{best}$, we then compute the equivalent set of non-overlapping lineages presented in this paper (Figs. 1–5 and Extended Data Fig. 4). We make sure the minimum number of nodes per non-overlapping lineage is at least $N_{min}$ by merging the small lineages to their respective phylogenetically closest lineages.

In simulations, we demonstrate a clear elbow that precisely identifies the optimal number of discrete lineages in the dataset (Supplementary Fig. 16). However, not all pathogens in real-world datasets will give clear elbows. This may be due to insufficient sampling intensity and the presence of lineages with only very small differences in fitness. In practice, increasing the number of distinct lineages will progressively lead to the identification of lineages with increasingly reduced fitness differences, with increasing risks of falsely dividing lineages into subpopulations between which no true differences exist.

For the pathogens presented in our study we found: SARS-CoV-2, 14 lineages; H3N2, 20 lineages; *B. pertussis*, 8 lineages; and *M. tuberculosis*, 12 lineages.

To compare the automatic lineages found with phylowave to those previously identified, we compute a contingency matrix $C$. Let $U$ be the partition of the isolates by phylowave and $V$ the partition based on the literature. Each element $C_{i,j}$ is the number of isolates in both clusters $u_i$ and $v_j$. In Fig. 2, we plot the matrix as a heat map, normalized by column $j$. We computed the ARI to measure the agreement between partitions, accounting for random clustering[21]. A value of 1 corresponds to perfect agreement with previously identified lineages, whereas a value of 0 would be expected if clusters were assigned at random.

We illustrate the impact of the timescale on lineage detection in Extended Data Fig. 8 for the global SARS-CoV-2 tree.

## Quantifying the fitness of each lineage

We developed a multinomial logistic model that takes into account the birth of lineages to fit the proportion of each lineage through time and quantify their fitness.

The proportion $p_{\cdot,t}$ of sequences at time $t$ from each lineage is computed as the number of nodes (internal and terminal) divided by the total number of nodes (internal and terminal) in the population at that time. This proportion $p_{\cdot,t}$ is modelled by:

$$p_{\cdot,t} = \text{softmax}(\log(\alpha_{\cdot}) + \beta_{\cdot}t)$$

Where $\alpha_{\cdot}$ is the vector of the intercept, denoting the initial relative prevalence of each lineage in the population and $\beta_{\cdot}$ the vector of the relative growth rates of each lineage. We assume that each lineage $i$ has a constant relative growth rate $\beta_i$ in the population, that is, each lineage has a constant relative fitness through time. We compute all the relative growth rates with reference to the oldest lineage.

We use a Laplace prior for the growth rate coefficient (where ~ denotes 'distributed as')[8]:

$$\beta_{\cdot} \sim \text{Laplace}(0, 1)$$

We take into account lineage birth by only allowing only $p_{i,t}$, the lineage $i$ proportion in the population at time $t$, to be non-negative after the MRCA of the lineage. Formally, this is done by parameterizing $\alpha_{\cdot}$ as follows. We divide the lineages into two types, either ancestral or

non-ancestral. An ancestral lineage is a lineage that is present at the beginning of the time series considered. The total number of ancestral lineages is noted $G$. For those lineages, we sample directly their starting proportions with prior:

$$\gamma_i \sim \text{simplex}(G); \text{ if } i \in \text{ancestors}$$

A non-ancestral lineage is a lineage that appears after some time—for example the Omicron variant. For those lineages, we assume that their starting frequency, at the time of emergence, is a function of the proportion of their parents in the population at that time. Thus we write:

$$\alpha_i = \gamma_i p_{j,t_{\text{MRCA}i}}; \text{ if } i \notin \text{ancestors}$$

Where $j$ is the parent lineage of lineage $i$, $p_{j,t_{\text{MRCA}i}}$ is the proportion of the parent lineage $j$ at the time of emergence $t_{\text{MRCA}i}$ of the offspring lineage $i$ and $\gamma_i$ is the share of the parent lineage that is becoming the new lineage. We sample $\gamma_i$ with a strong prior because we expect that the starting proportion of new lineages should be small:

$$\gamma_i \sim \text{beta}(1, 99); \text{ if } i \notin \text{ancestors}$$

Finally, we update the parent $j$ proportion as follows:

$$p_{j,t_{\text{MRCA}i}+\delta} = (1 - \gamma_i)p_{j,t_{\text{MRCA}i}}$$

Although this parameterization is more complex than the previous efforts using a similar model[8], it enables us to take into account that lineages appear through time, which makes the model more biologically relevant (for example, by not estimating the proportion of Omicron in the population in 2020). We chose to parameterize the starting proportions of the new lineages as a function of the proportion of their parents so that the model is biologically sound, that is, the starting proportion of a new lineage cannot be greater than the one of its parent, and the starting proportions are constrained by the proportion of the parents. The parameters make the model statistically easier to fit.

We use a multinomial likelihood to fit the count of sequences per lineage through time $y_{\cdot,t}$:

$$y_{\cdot,t} \sim \text{multinomial}\left(\sum_i y_{i,t}, p_{\cdot,t}\right)$$

We further computed the inferred real-time growth rate (that is, the fitness) $r_i(t)$ of each lineage $i$ in the population (Fig. 3e–h), to control for the varying presence of all circulating lineages through time. Indeed, although our model estimates a constant fitness parameter for each lineage, their actual fitness through time depends on what other lineages are circulating at that time.

$$r_i(t) = p_{i,t} \sum_{\substack{j \in \text{lineages,} \\ j \neq i}} p_{j,t}(\beta_i - \beta_j)$$

These results are more useful than the usual presentation of the parameters, which by default display the relative fitness compared with the ancestral lineage, in this case 19A (the lineage that includes the first SARS-CoV-2 sequences isolated in Wuhan, China).

The model was implemented in Stan, using the cmdstanr package[61]. We ran this model on three independent chains with 1,000 iterations and 50% burn-in for each pathogen. We used 2.5% and 97.5% quantiles from the resulting posterior distributions for the 95% credible intervals of the parameters.

We fit the counts per lineage in windows of 1 month for SARS-CoV-2, 0.2 year for H3N2, 1 year for *B. pertussis* and 20 years for *M. tuberculosis*, with $t$ counted in years for all pathogens.

### Defining mutations of each lineage

We explored whether specific changes in the genomes were linked to lineage fitness by identifying lineage-defining mutations. We defined such mutations as follows: (i) mutations that are present in more than 80% of the nodes in that lineage; (ii) mutations that are not present in the set of defining mutations of the ancestral lineage.

For all pathogens, we reconstructed the mutations at each tree node using the ancestral state reconstruction implemented in the library ape. To maximize the correct assignment of nodes, we consider only nodes for which the probability of the state was >0.9. Mutations were then classified as synonymous, non-synonymous or extragenic. For *M. tuberculosis* and *B. pertussis*, we also classified each mutation by functional category[35,36].

We computed the density of lineage-defining mutations along the SARS-CoV-2 full genome and H3N2 HA polyprotein with a kernel density estimate (Fig. 4e,f). We used a Gaussian kernel with a bandwidth of 50 base pairs for SARS-CoV-2, and a bandwidth of 2.5 amino acids for H3N2. For *B. pertussis* and *M. tuberculosis*, we plot for each mutation the maximum proportion of that mutation that is present in the set of groups considered.

To assess the functional relevance of the mutations identified for each pathogen (Supplementary Tables 6–9), we compared them with the literature.

For SARS-CoV-2, we matched the amino acid substitution we found with the ones that have been analysed previously[8]. The authors analysed 6.4 million genomes up to 20 January 2022 and estimated the fitness effect of 2,904 substitutions. Although our global dataset encompasses a longer period (up to 3 April 2023), 83% ($n = 161$) of the lineage-defining alterations had been analysed previously[8]. Our approach was able to recover all of the top 55 fittest alterations described previously[8]. Among the top 100 fittest alterations, our approach recovered 86% of them. The substitution missed by our method are mainly linked to small subclades of variants, and they seem to have spread in those clades only (for example, in subclades of Delta 21I: ORF1a(T3750I) and ORF1b(R188Q)). One substitution was missed because of a lack of certainty in the ancestral state reconstruction around the root of the Omicron sublineages: S(T376A). Of the alterations that were estimated to have no fitness increase, we found seven (among 2,331 analysed previously[8]). These alterations include S(D614G) and ORF1b(P314L), two alterations that are linked to an early lineage that eventually got fixed in the whole population. We also found ORF8(G8*) and S(G252V), which defined our lineage 1 (XBB1*), and ORF1a(L3829F), S(N460K) and ORF1b(M1156I), which defined our lineage 4 (22E/BQ.1). These substitutions were analysed by previously[8] but were present only at a very small frequency in their dataset, as they are mainly linked to the emergence of recent variants, which emerged after their study.

For H3N2, we computed the proportion of positions that are lineage defining in each HA polyprotein subunit and the antigenic sites[32,33]. A position is lineage defining if it has at least one amino acid (AA) substitution that is lineage defining. The proportion is computed as follows:

$$\pi_L = \frac{\text{Number of positions that are lineage defining in } L}{\text{Number of positions that are mutated in } L}$$

Where $L$ is the set of positions to be analysed (subunits or antigenic sites). We found that the Koel sites[33] had the highest proportion of lineage-defining mutations, with 86% of positions ($n = 6$, out of 7) being recovered by phylowave. Specifically, the key positions 156, 159 and 193, defining multiple clades, were recovered. We also recovered positions that defined ancestral lineages, namely positions 145, 158 and 189. Furthermore, position 155 was not picked up by our method because it did not have any variability in our dataset. Indeed, this position was found to be linked to major antigenic changes in the 1960s and 1970s; this period does not overlap with our study period (2005–2023).

For the bacteria *B. pertussis* and *M. tuberculosis* we use a similar metric, by grouping mutations according to gene functional categories. We compute:

$$\pi_F = \frac{\text{Number of AA substitutions that are lineage defining in } F}{\text{Number of AA substitutions in } F}$$

Where $F$ is the gene functional category that we considered[35,36]. As a sensitivity analysis, we also replicated this computation on synonymous nucleotide changes, as we expect these mutations to be neutral, and therefore not linked to any particular functional category (Supplementary Fig. 17). We found that, indeed, there was no particular functional category that had significantly more lineage-defining synonymous mutations than others, for both bacteria.

To further check our findings visually, we plotted the lineage-defining mutations for each pathogen next to their phylogenetic trees (Supplementary Figs. 9–12). To make sure the figures were interpretable, we plotted only the mutations in the spike protein for SARS-CoV-2 (Supplementary Fig. 9), the HA1 subunit for H3N2 (Supplementary Fig. 10) and the mutations defining lineages 1 and 2 for *M. tuberculosis* (Supplementary Fig. 12). For *B. pertussis*, we plotted all mutations (amino acid substitutions and promoter mutations) (Supplementary Fig. 11).

### Robustness to sampling strategies

To demonstrate the robustness to sampling biases over time, we conducted a sensitivity analysis using the global SARS-CoV-2 dataset. We selected two random sets of 150 sequences from the 3,129 sequences in our full dataset. We selected them either uniformly through time or in a temporally uneven manner. To do so, we divided the sequences in 15 time windows of equal length (79 days). For the uniform sampling, we included ten randomly selected sequences per time bin. For the biased sampling, we included the following number of sequences per bin (see inset on Fig. 5b): windows 1 and 2, 1 sequence per bin; windows 3–5, 2 sequences per bin; windows 6–9, 25 sequences per bin; windows 10–15, 7 sequences per bin.

After selecting the sequences, we pruned from the tree the ones that were not selected. We then performed the same analysis as described above. We also compared the groups found.

### Analysis of time to detection

We explored how fast after emergence phylowave was able to detect lineages. To do this we truncated our full global SARS-CoV-2 dataset every two weeks. Overall, we obtained 81 datasets. Two examples of the index dynamics for data censored on 2021.26 and 2022.50 are presented in Extended Data Fig. 10. We then reran the detection algorithm on each dataset. To obtain the best set of lineages automatically for each dataset, we chose the set for which the log deviance explained did not increase by more than 0.01%.

### Reporting summary

Further information on research design is available in the Nature Portfolio Reporting Summary linked to this article.

## Data availability

All *B. pertussis* sequences generated for this study were deposited in ENA, with accession numbers and metadata attached for each individual sequence available in Supplementary Table 10. All sequences and metadata used in this study, including the reference sequences, are listed in Supplementary Tables 10–13. All sequences are publicly available online on GenBank and ENA (*B. pertussis*, *M. tuberculosis*[20]) or GISAID (H3N2, SARS-CoV-2). Supplementary Tables 10–13 are also available at Zenodo (https://zenodo.org/records/13952222)[62].

## Code availability

Code to replicate the main analyses of this paper is available at Zenodo (https://zenodo.org/records/13952222)[62]. General guidelines to use phylowave and a step-by-step example are included in Supplementary Information sections 3 and 4.

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

**Acknowledgements** We thank C. Collins, M. O'Driscoll, A. T. Huang and T. Bedford for discussions and feedback and all the contributors to GISAID for sharing their data. This work was supported financially by the European Research Council (no. 804744 to N.L. and H.S.) and the UK Research and Innovation, Medical Research Council (no. MR/Z504373/1 to H.S.). The National Reference Center for Whooping Cough and Other Bordetella Infections receives support from Institut Pasteur and Public Health France (Santé publique France).

**Author contributions** Conceptualization: N.L., J.P. and H.S. Method development and modelling analysis: N.L., supported by L.D., J.P. and H.S. Isolate and genomic data collection: N.L., S.B. and V.B. Supervision: J.P. and H.S. Writing—original draft: N.L. Writing—review and editing: N.L., L.D., V.B., S.B., J.P. and H.S. All authors provided input to the manuscript and reviewed the final version.

**Competing interests** The authors declare no competing interests.

**Additional information**
**Correspondence and requests for materials** should be addressed to Noémie Lefrancq.

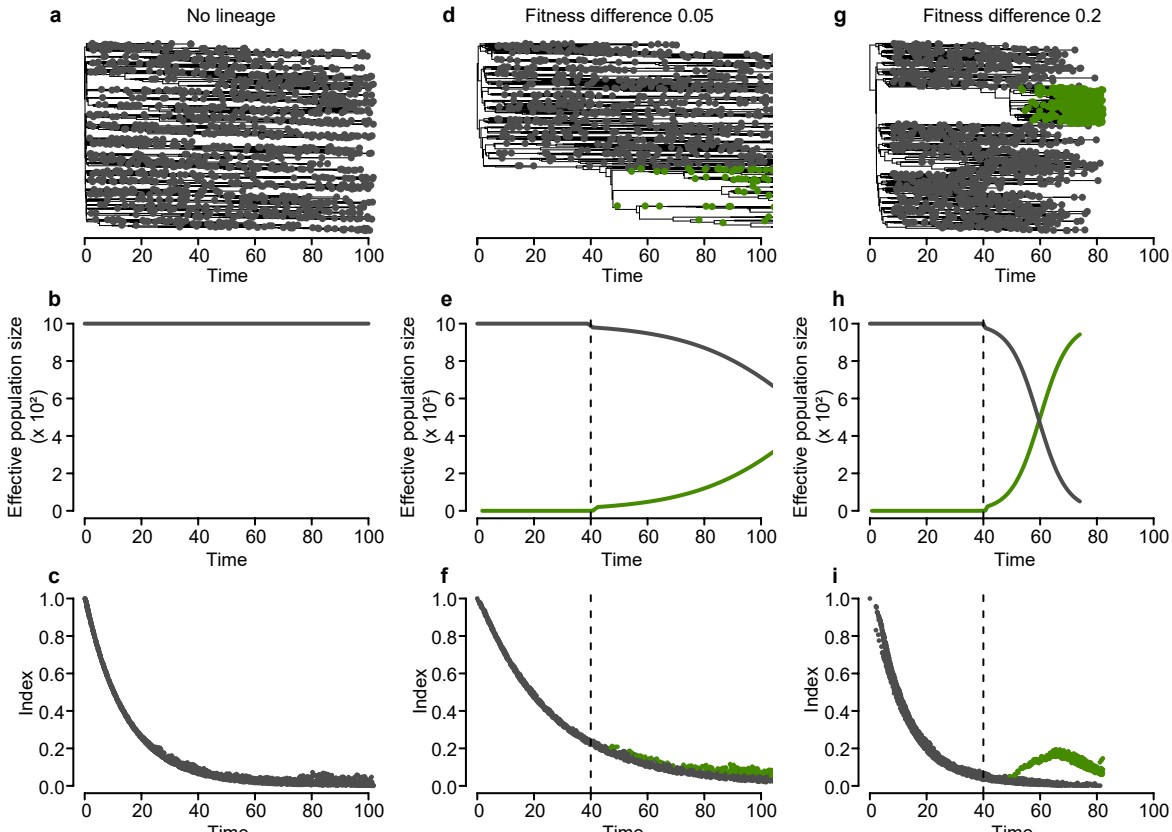

**Extended Data Fig. 1 | Example of simulated dynamics for different fitness advantages.** We present examples of simulations in the case of no emerging lineages (**a**–**c**) and an emerging lineage with a fitness difference of 0.05 per time unit (**d**–**f**), or 0.2 per time unit (**g**–**i**). For each condition, we present a simulated tree, the effective population size for each lineage and the index of each node in the simulated tree. Colours denote each population: the background (grey) and the emerging lineage (green). The dashed line represents the time at which the emerging lineage was introduced (T = 40).

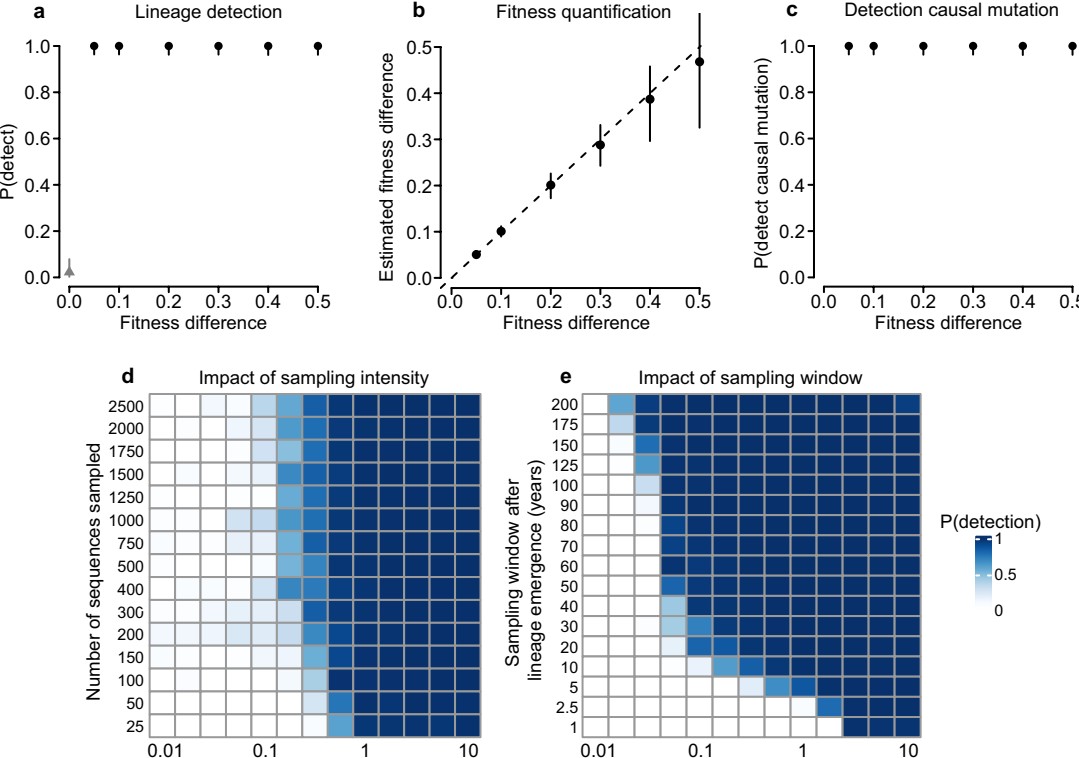

**Extended Data Fig. 2 | Results of the simulation study to assess the ability of phylowave to detect emerging lineages.** (**a**–**c**) For each fitness value, we simulated a full lineage replacement (until the emerging lineage represents 95% of the population) and extracted 100 trees of 1500 tips each. We plot the proportion of times we detect the emerging lineage (**a**), the estimated fitness difference, using our multinomial logistic model (**b**) and the proportion of times we detect the causal mutation (**c**). Dots and bars in (**a**) and (**c**) denote the median and 95% binomial confidence intervals. The grey triangle in (**a**) denotes the case where no emerging lineages were simulated. Dots and bars in (**b**) denote the median and 95% credible interval of the model posterior. Details on the simulation studies can be found in Supplementary Text 2. (**d**–**e**) Ability of phylowave to detect an emerging lineage in datasets of different sizes (**d**) or sampling window post lineage emergence (**e**). In (**d**) we fix the sampling window after lineage emergence to 10 years. In (**e**) we fix the size of the datasets to 1,500 sequences. For each scenario, 20 trees were simulated. Details on the simulation studies can be found in Supplementary Text 2.

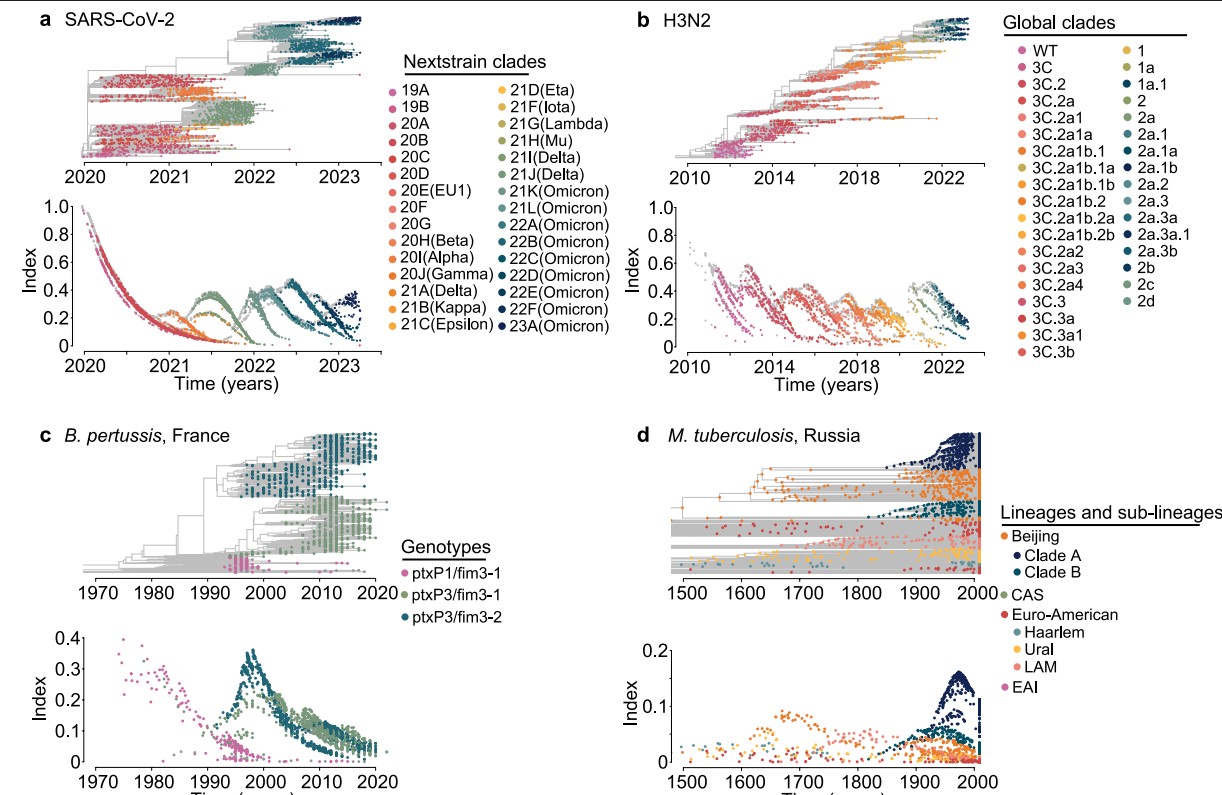

**Extended Data Fig. 3 | Index dynamics coloured by known lineages.** Similar to Fig. 1, for SARS-CoV-2 (**a**), H3N2 (**b**), *B. pertussis* (**c**) and *M. tuberculosis* (**d**), we present the index dynamics computed at each node (terminal or internal). Here colours represent the different known clades, genotypes or lineages (see legend on the side). For *M. tuberculosis*, LAM denotes the Latin American–Mediterranean lineage, EAI denotes the East African Indian lineage and CAS denotes the Central Asian Strain lineage.

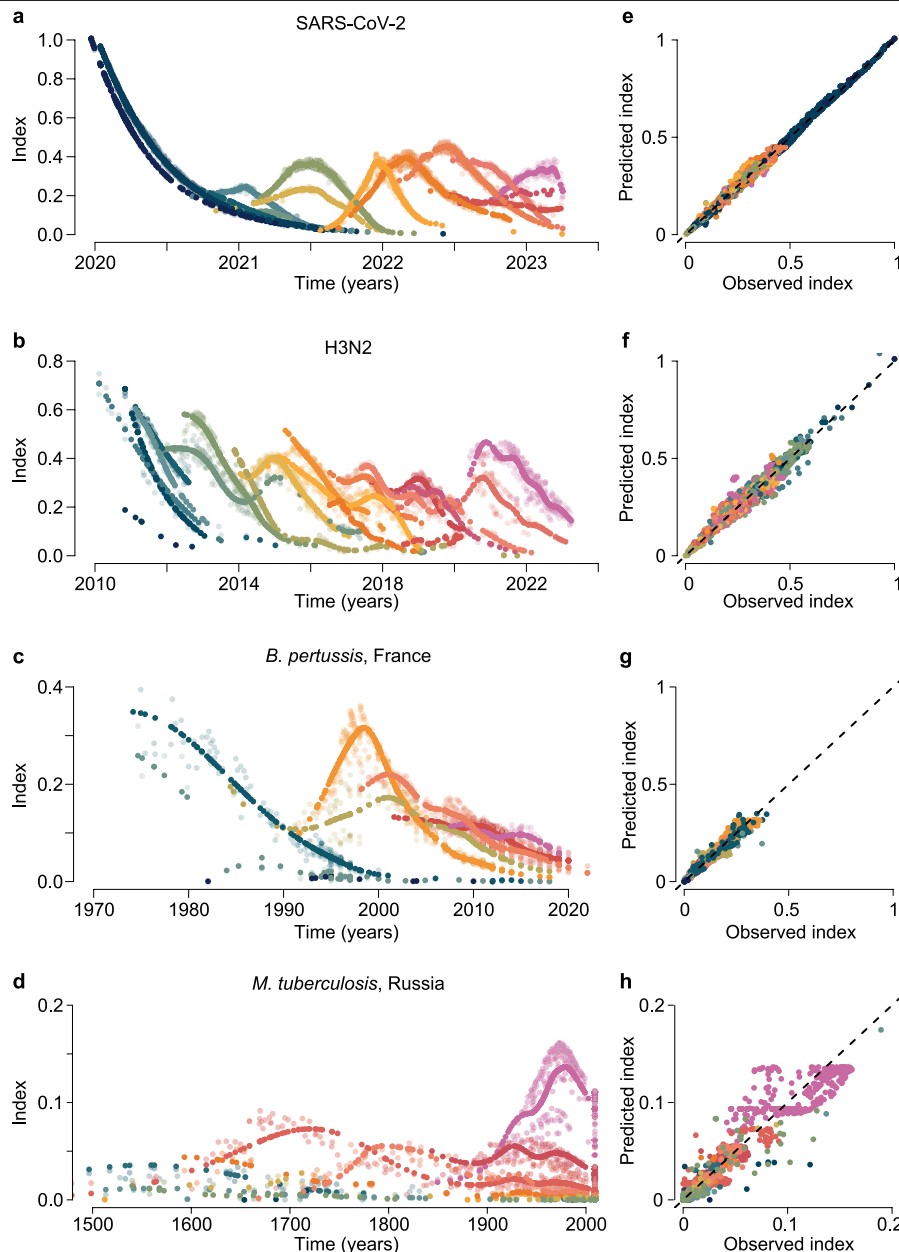

**Extended Data Fig. 4 | Lineage detection based on index dynamics for each pathogen. (a–d)** For each pathogen we present model fits of the index dynamics using the best set of lineages. Solid dots represent the model prediction. Shaded dots represent the data. **(e–h)** Predicted versus observed index.

The dashed lines denote identity lines. For each pathogen, colours represent the different lineages identified by their different index dynamics (same colours as in Figs. 1–4).

**a** SARS-CoV-2

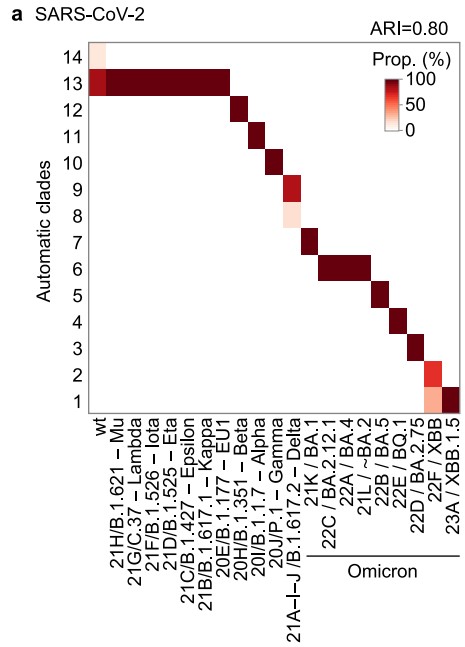

**b** H3N2

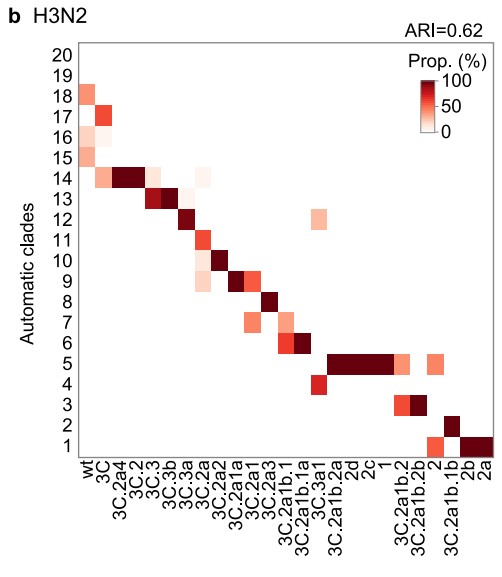

**c** *B. pertussis*, France

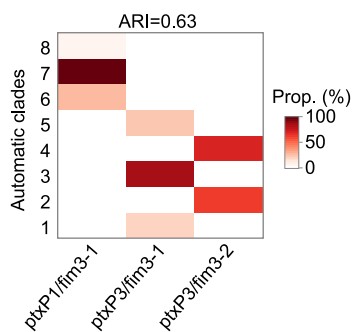

**d** *M. tuberculosis*, Russia

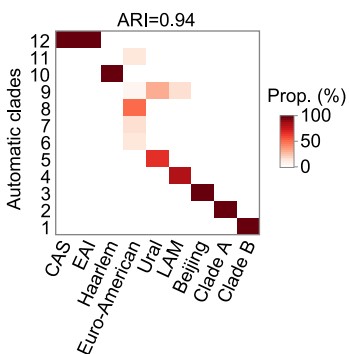

**Extended Data Fig. 5 | Heat maps comparing the identified lineages to the known population composition.** For SARS-CoV-2 (**a**), H3N2 (**b**), *B. pertussis* (**c**) and *M. tuberculosis* (**d**), we present a heat map comparing the known population structure (*x* axis) to the automatic clades found by our phylowave (*y* axis). Darker colours represent more agreement between both classifications. Contingency tables are presented in Supplementary Tables 2–5.

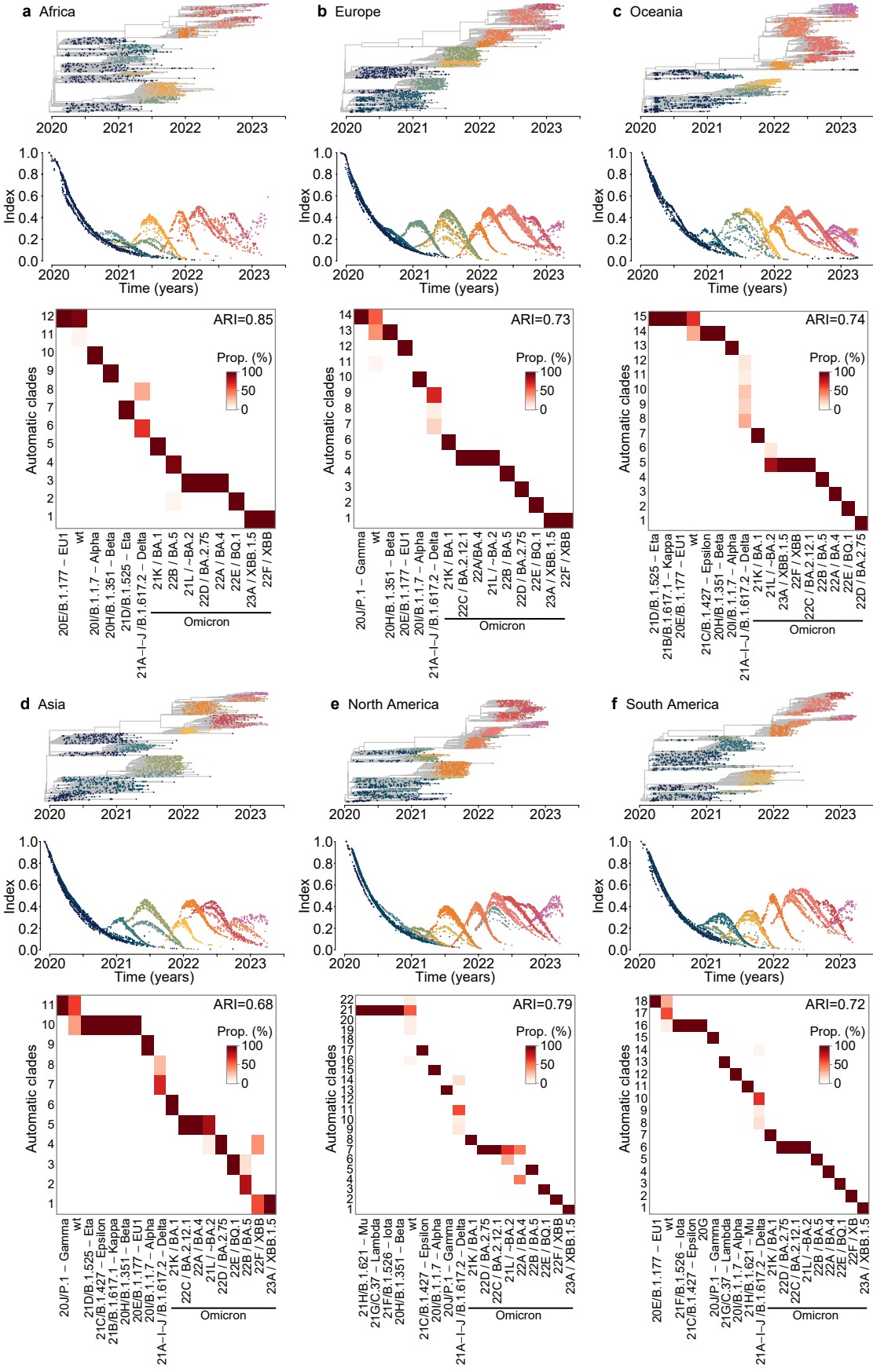

**Extended Data Fig. 6 |** See next page for caption.

**Extended Data Fig. 6 | SARS-CoV-2 index dynamics and lineages identified across continents.** We present the index dynamics computed at each node (terminal or internal) for datasets of SARS-CoV-2 isolated in Africa (**a**), Europe (**b**), Oceania (**c**), Asia (**d**), North America (**e**) and South America (**f**). The colours of the dots represent the different lineages identified by their different index dynamics. Time-resolved phylogenies for each continent were obtained from NextStrain, accessed on 14 April 2023[37]. For each continent we also present a heat map comparing the known clades identified by NextStrain (*x* axis) to the automatic clades found by phylowave (*y* axis). Darker colours represent more agreement between both namings.

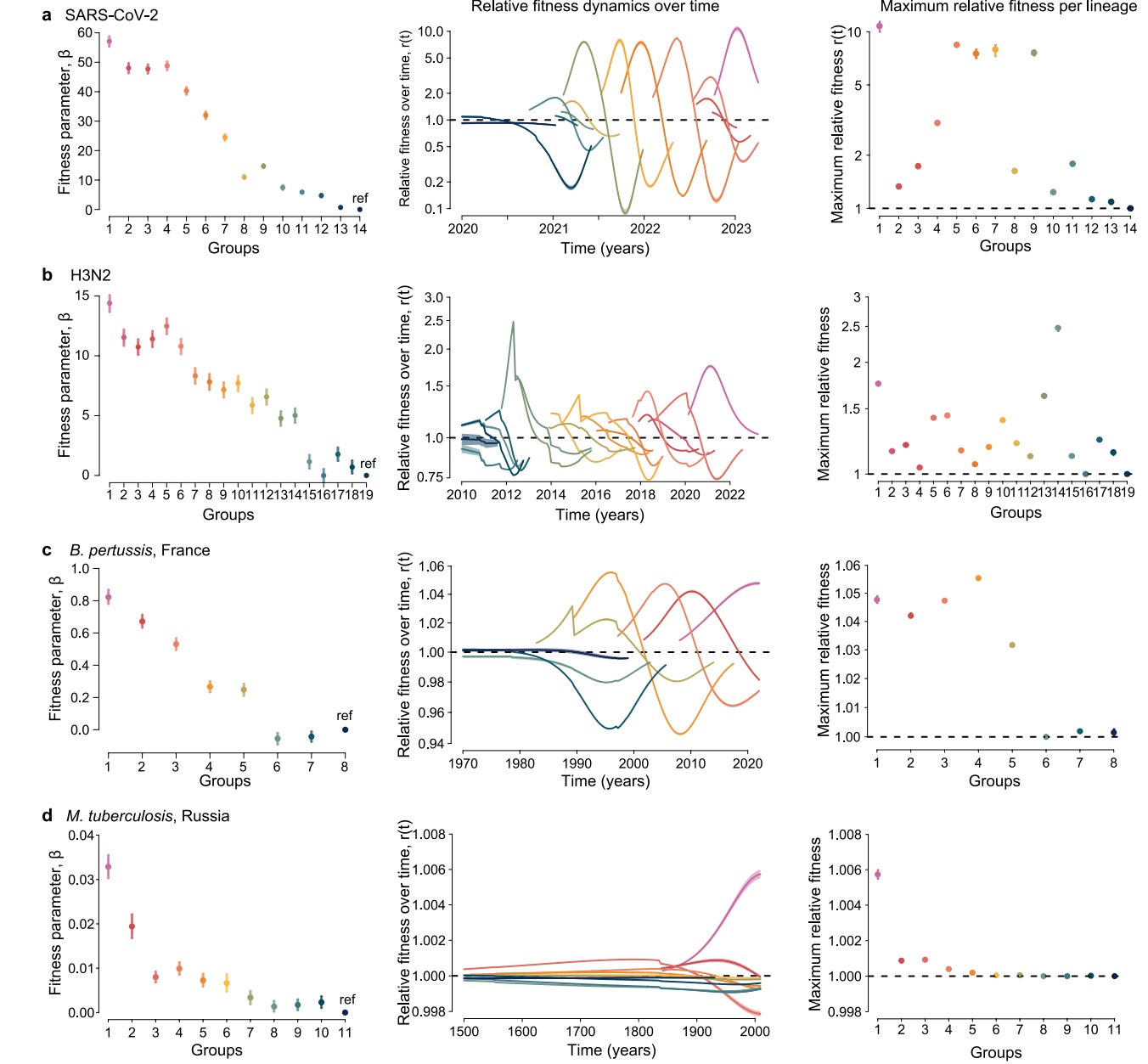

**Extended Data Fig. 7 | Fitness estimates for all pathogen lineages.** For SARS-CoV-2 (**a**), H3N2 (**b**), *B. pertussis* (**c**) and *M. tuberculosis* (**d**) we present the estimated fitness of each of their lineages. From left to right: Fitness parameter $\beta$ for each lineage; Relative fitness dynamics overtime $r(t)$; maximum relative fitness per lineage. Dots represent median estimates for each lineage, bars denote 95% credible interval of the posterior. Lines and shaded areas represent the median and 95% credible interval of the posterior. Colours represent the different lineages identified for each pathogen.

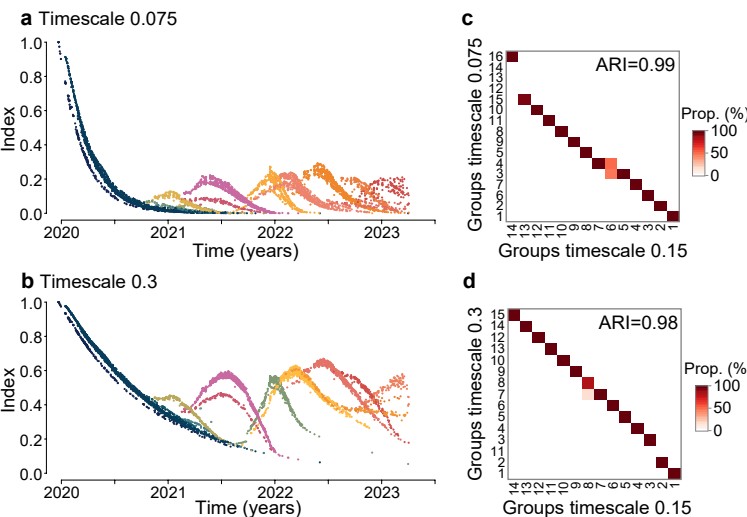

**Extended Data Fig. 8 | Robustness of phylowave to the choice of timescale.** We show phylowave is robust to the choice of the timescale (governing the weight distribution used in the index computation). (a–b) Index dynamics computed on the global SARS-CoV-2 tree, with either a timescale of 0.075 (a) or 0.3 (b). The timescale used in the main analysis is 0.15. A smaller timescale focused more on recent population dynamics, a larger timescale focused more on the past evolution. Colours represent the lineages identified with our algorithm on those dynamics. (**c–d**) We compare the lineages identified with those timescales (*y* axis) to the lineages presented throughout this study, with a timescale of 0.15 (*x* axis). Darker colours represent more agreement between both namings. Overall, we find minimal differences in the lineages detected. Our results for SARS-CoV-2 are robust to the exact chosen timescale.

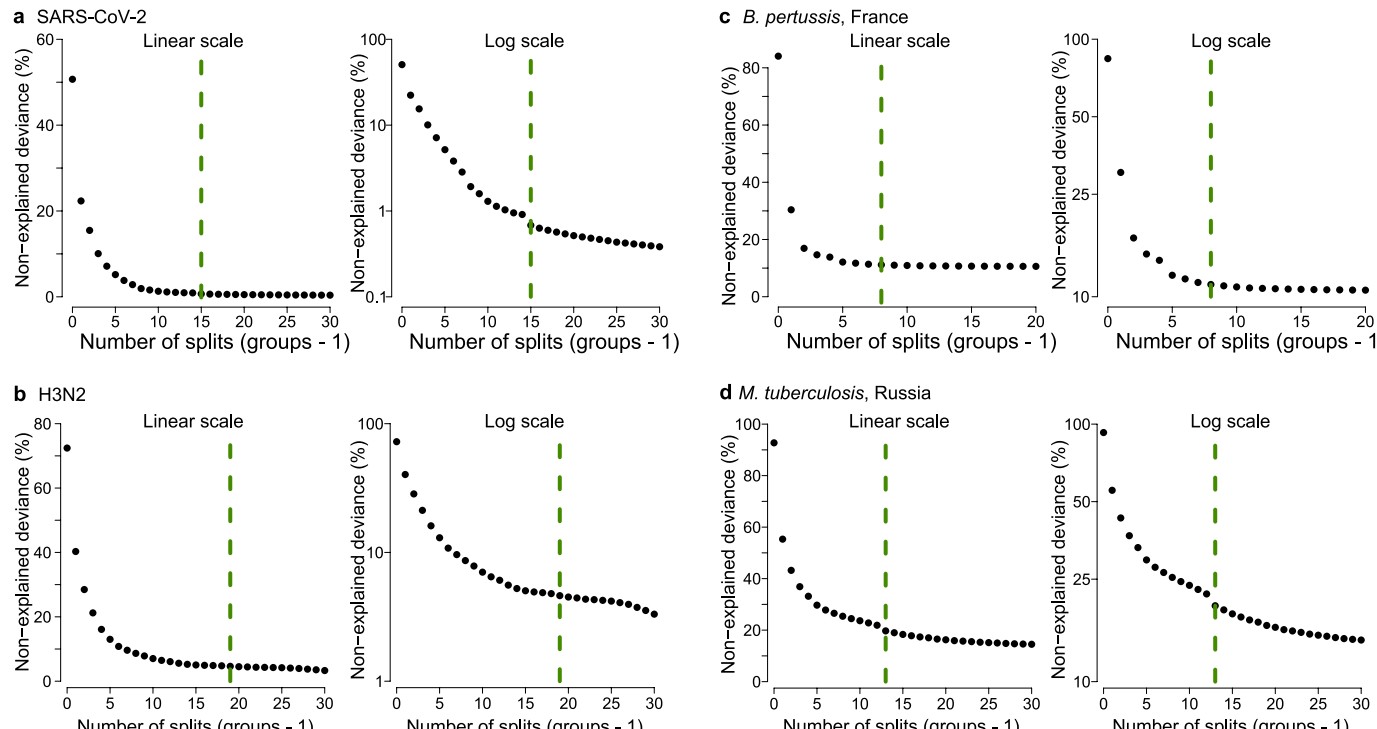

**Extended Data Fig. 9 | Non-explained deviance as a function of the number of groups in the lineage detection algorithm.** For SARS-CoV-2 (**a**), H3N2 (**b**), *B. pertussis* (**c**) and *M. tuberculosis* (**d**), we plot the proportion of non-explained deviance by the models with different numbers of groups. Dashed lines represent the number of groups chosen. We plot the proportion both on a linear scale (left) and log scale (right). The log scale enables a more precise appreciation of the number of groups at which the deviance explained does not increase substantially anymore.

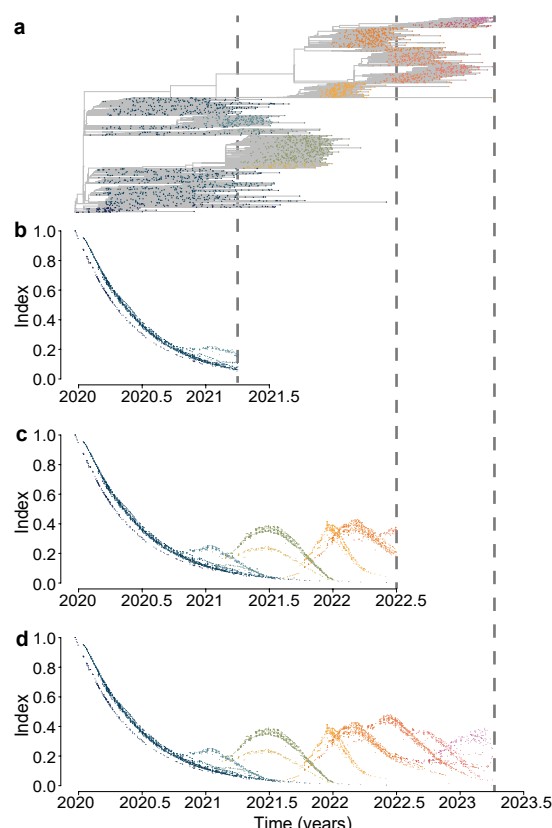

**Extended Data Fig. 10 | Example of index dynamics on time censored global SARS-CoV-2 datasets.** (**a**) Global SARS-CoV-2 time-resolved phylogenetic tree, same as on Figs. 1 and 2. Dots denote terminal nodes only. (**b**–**c**) Index computed on censored datasets, on either 2021.26 (**b**) or 2022.5 (**c**). (**d**) Uncensored index dynamics. When censoring a dataset we prune all isolates not selected, effectively removing internal nodes as well as terminal nodes. This explains the slightly different dynamics observed near the censoring date. Colours represent the lineages automatically found by phylowave (same as Figs. 1 and 2).

# Reporting Summary

## Statistics

For all statistical analyses, confirm that the following items are present in the figure legend, table legend, main text, or Methods section.

| n/a | Confirmed | |
|---|---|---|
| ☐ | ☒ | The exact sample size (*n*) for each experimental group/condition, given as a discrete number and unit of measurement |
| ☐ | ☒ | A statement on whether measurements were taken from distinct samples or whether the same sample was measured repeatedly |
| ☐ | ☒ | The statistical test(s) used AND whether they are one- or two-sided *Only common tests should be described solely by name; describe more complex techniques in the Methods section.* |
| ☐ | ☒ | A description of all covariates tested |
| ☐ | ☒ | A description of any assumptions or corrections, such as tests of normality and adjustment for multiple comparisons |
| ☐ | ☒ | A full description of the statistical parameters including central tendency (e.g. means) or other basic estimates (e.g. regression coefficient) AND variation (e.g. standard deviation) or associated estimates of uncertainty (e.g. confidence intervals) |
| ☒ | ☐ | For null hypothesis testing, the test statistic (e.g. *F*, *t*, *r*) with confidence intervals, effect sizes, degrees of freedom and *P* value noted *Give P values as exact values whenever suitable.* |
| ☐ | ☒ | For Bayesian analysis, information on the choice of priors and Markov chain Monte Carlo settings |
| ☒ | ☐ | For hierarchical and complex designs, identification of the appropriate level for tests and full reporting of outcomes |
| ☒ | ☐ | Estimates of effect sizes (e.g. Cohen's *d*, Pearson's *r*), indicating how they were calculated |

*Our web collection on statistics for biologists contains articles on many of the points above.*

## Software and code

Policy information about availability of computer code

| | |
|---|---|
| Data collection | For each pathogen, we compiled a dataset to investigate the changes in the population composition. For SARS-CoV-2 and Influenza H3N2, we extracted the datasets from the publicly available NextStrain37 timed-resolved phylogenies accessed on 14 April 2023. These datasets are sub-samples from all publicly available sequences in GISAID, to represent the diversity as much as possible (we used the 'all-time' dataset for SARS-CoV-2 and the '12y' one for H3N2). In all, we have 3129 whole genome SARS-CoV-2 sequences sampled from 26 December 2019 to 3 April 2023, and 1476 Influenza H3N2 Hemagglutinin (HA) sequences from 1 January 2005 to 3 April 2023 (Supplementary Tables 10-11). For B. pertussis, we used 1248 sequences from 1953 to 2022, collected by the National Reference Center (NRC) for Whooping Cough and Other Bordetella Infections in France (Supplementary Table 12). This dataset is composed of 1023 sequences previously published3,38–41 and 225 newly sequenced isolates. The new isolates have been sequenced with the same methods as previously described3. This dataset is representative of the B. pertussis diversity in France as the NRC is receiving isolates from 42 sentinelle hospitals throughout France. For M. tuberculosis, we used 997 previously published sequences, isolated in 2008-2010 in Samara, Russia20. This dataset is also representative of M. tuberculosis sequence diversity at that location as isolates were prospectively collected from individual patients living in the region and representative of the entire population (Supplementary Table 13). |
| Data analysis | To create the multi-sequence alignment for each pathogen, the following softwares and algorithms were used: MAFFT (v7.309), Cutadapt (v3.4), FastQC (v0.11.9), BWA-MEM (v0.7.17), GATK (v4.2.0.0), PHASTER, Gubbins (v3.3.0). <br> To reconstruct the time-resolved phylogenetic trees, the following softwares and algorithms were used: IQ-tree (v2.1.0), BEAST (v1.10.4), BEAGLE library (v4.0.0), Tracer (v1.7.2), BactDating (v1.1). <br> For the main code analysis, we wrote the code in R v4.1.2, and used the package mgcv v1.8 to build the detection algorithm. <br> We compared phylowave to fastbaps v1.0.8 and treestructure. |

For manuscripts utilizing custom algorithms or software that are central to the research but not yet described in published literature, software must be made available to editors and reviewers. We strongly encourage code deposition in a community repository (e.g. GitHub). See the Nature Portfolio guidelines for submitting code & software for further information.

# Data

Policy information about availability of data

All manuscripts must include a data availability statement. This statement should provide the following information, where applicable:
- Accession codes, unique identifiers, or web links for publicly available datasets
- A description of any restrictions on data availability
- For clinical datasets or third party data, please ensure that the statement adheres to our policy

Data availability statement:
All B. pertussis sequences generated for this study were deposited in ENA, with accession numbers and metadata attached for each individual sequence available in Supplementary Table 10. All sequences and metadata used in this study, including the reference sequences, are listed in Supplementary Tables 10-13. All sequences are publicly available online on GenBank and ENA (B. pertussis, M. tuberculosis20) or GISAID (H3N2, SARS-CoV-2). The Supplementary Tables 10-13 are also available online in the repository: https://zenodo.org/records/13952222 [Ref: 62].

# Research involving human participants, their data, or biological material

Policy information about studies with human participants or human data. See also policy information about sex, gender (identity/presentation), and sexual orientation and race, ethnicity and racism.

| | |
|---|---|
| Reporting on sex and gender | NA |
| Reporting on race, ethnicity, or other socially relevant groupings | NA |
| Population characteristics | NA |
| Recruitment | NA |
| Ethics oversight | Data on SARS-CoV-2, H3N2 and M. tuberculosis were previously published. B. pertussis data was obtained by the French National Reference Centre for Whooping Cough and Other Bordetella infections, which is authorised to conduct sample collection. |

Note that full information on the approval of the study protocol must also be provided in the manuscript.

# Field-specific reporting

Please select the one below that is the best fit for your research. If you are not sure, read the appropriate sections before making your selection.

☐ Life sciences    ☐ Behavioural & social sciences    ☒ Ecological, evolutionary & environmental sciences

For a reference copy of the document with all sections, see nature.com/documents/nr-reporting-summary-flat.pdf

# Ecological, evolutionary & environmental sciences study design

All studies must disclose on these points even when the disclosure is negative.

| | |
|---|---|
| Study description | For each pathogen, we compiled a dataset to investigate the changes in population composition with our novel framework. |
| Research sample | Whole-genomes of SARS-CoV-2, Bordetella pertussis and Mycobacterium tuberculosis, and HA gene of H3N2. |
| Sampling strategy | For each pathogen studied, we gathered a dataset that is representative of that pathogen's genetic diversity. |
| Data collection | Data on SARS-CoV-2, H3N2 and M. tuberculosis were previously published.B. pertussis data was obtained by the French National Reference Centre for Whooping Cough and Other Bordetella infections. |
| Timing and spatial scale | SARS-CoV-2: 2019-2023, worldwide,<br>H3N2: 2005-2023, worldwide,<br>B. pertussis: 1953-2022, France<br>M. tuberculosis: 2008-2010, Russia |

| Data exclusions | NA |
|---|---|
| Reproducibility | We only used natural experiment in this study (pathogen circulating in human population) - analysis is reproducible using codes |
| Randomization | We only used natural experiment in this study (pathogen circulating in human population) |
| Blinding | We only used natural experiment in this study (pathogen circulating in human population) |

Did the study involve field work?  ☐ Yes  ☒ No

# Reporting for specific materials, systems and methods

We require information from authors about some types of materials, experimental systems and methods used in many studies. Here, indicate whether each material, system or method listed is relevant to your study. If you are not sure if a list item applies to your research, read the appropriate section before selecting a response.

## Materials & experimental systems

| n/a | Involved in the study |
|---|---|
| ☒ | ☐ Antibodies |
| ☒ | ☐ Eukaryotic cell lines |
| ☒ | ☐ Palaeontology and archaeology |
| ☒ | ☐ Animals and other organisms |
| ☒ | ☐ Clinical data |
| ☒ | ☐ Dual use research of concern |
| ☒ | ☐ Plants |

## Methods

| n/a | Involved in the study |
|---|---|
| ☒ | ☐ ChIP-seq |
| ☒ | ☐ Flow cytometry |
| ☒ | ☐ MRI-based neuroimaging |

## Plants

| Seed stocks | NA |
|---|---|
| Novel plant genotypes | NA |
| Authentication | NA |

