## [Peer Review file · Nature]

Learning the fitness dynamics of pathogens from phylogenies

Corresponding Author: Dr Noémie Lefrancq

Version 0:

Reviewer comments:

Referee #1

(Remarks to the Author)

This paper presents an intriguing looking method for identifying positively-selected lineages within pathogen populations. There are applications to important and topical pathogens. Some nice, although hardly revolutionary stories are told and it does present a nice overview of the evidence of fitness changes in these pathogens.

My main issue with the paper is that it presents a new method but does not test the method in a situation where the ground truth is known, namely simulated data, where a species evolves with occasional fitness-changing mutations on an otherwise neutrally evolving background. If the method is working, then it should be able to identify the lineages with reasonable precision and should include the causal mutation within the set of lineage defining mutations it identifies.

Being able to recreate previously identified lineages automatically - which is the main validation criterion given by the authors - is not in itself all that impressive. There are other automatic methods out there, as cited by the authors (such as fastBAPS), even if they make slightly different assumptions. And it might be that all of the methods are identifying similar signals, but they are NOT actually signals of selection, but rather simply the same stochastic events. There certainly is not objective evidence that their method is actually better. The robustness to sub-sampling is reassuring but similarly does not actually prove that the method is picking up selection signals as claimed.

Referee #2

(Remarks to the Author)

The paper of Lefrancq et al tackles a very important problem, of inferring pathogen lineages that are high fitness and spreading rapidly. The subject is timely in the context of SARS-CoV-2 but not only; and the authors go on to analyze data from four major pathogens, which is laudable. The paper is overall well written and clear, but there are some issues (see below). I am less of an expert on bacterial genomes and thus most of my comments are on the viral data analyzed.

I have the following comments:

1. I liked the idea of agnostic detection of lineages, and I agree with the authors on disadvantages of previous approaches, and especially the need to rely on previous notation of lineages (e.g., PANGO lineages for SARS-CoV-2). However, the use of multinomial logistic regression could also in principle be performed agnostically over all (or some) clades in the tree, similar to what the authors perform (but using a different index). My major comment if so, what would a comparison between these two approaches yield? What advantage (if) is there of the index approach?
2. The authors highly rely on the phylogeny. At the best of times, it might be better to take into account tree topology uncertainty; and with SARS-CoV-2 (or any pathogen sampled over short time scales), this might become even more profound (see Morel et al. MBE 2021). Could the authors address this shortcoming?
3. The authors compare their approach to the Obermeyer et al. paper and very briefly write that their "screening recovered the fittest mutations". More info is necessary here. What proportion? Which mutations are missed out? Which are newly inferred, and can the authors discuss why the discrepancy (as much as possible). Same goes for the Influenza analysis where there is very concrete evidence on antigenic sites and what they mean.

4. Similar to the above, which known lineages are detected, which are missed? Which non-previously-defined lineages are inferred and why?
5. General comments: while the figures are aesthetically appealing, they are all very hard to follow. First, the figures are tiny as are font sizes. Second, there is a lot of information in each panel that is not always explained. Some examples are given but this is rife across all figures:
 - a. Fig. 2 the ARI panels on the left, x-axis labels too small to see. The trees aligned against each other are appealing, but the authors need to explicitly write in the text which lineages are concordant, which are not, and why. It would be good to tie this to the ARI displayed
 - b. I have no idea what figure 4I is presenting
 - c. Figure 4A-D shows lineage, but this is meaningless unless some biological context is given for all lineages and discussed (comment 4).
 - d. in the ARI plots, in some cases it seems very low concordance, can the authors comment on this.
6. If I understand correctly, Figure 3E-H shows a pattern of initial negative fitness that becomes positive fitness for almost all lineages found. Is this a biological feature (changing antigenic landscape) or is this is an issue with sample size increasing? Why is this different across the four pathogens (e.g., for influenza seems much less prominent)?

Minor comments:

1. Line 63, the authors argue that some methods are susceptible to sampling biases, this is very true for the author's approach as well.
2. How much does the method rely on the sampling strategies employed by NextStrain. If the authors were to perform sampling on their own, how would their method fair?
3. While I understood the logic behind the index via figure 1, I had a hard time understanding the index defined in eq.1. Why is the summation over d rather than over nodes? Why are we summing over a distribution over and over?
4. Does evolutionary time (below figure 1) correspond to branch lengths?
5. How can the bandwidth b equal zero?
6. Figure S18 – I can't really see an "elbow" – which means the choice of the number of lineages is rather arbitrary.
7. Why are lineage defining mutation not inferred directly from the tree, as the mutations occurring on the lineage leading to the ancestral node?

Referee #3

(Remarks to the Author)

The paper by Lefrancq et al is about analysing phylogenetic trees created from DNA (or RNA) sequences of viruses and bacteria, which have been collected from a host population (e.g. humans) over a time period of typically some years. They use four real exemplar situations - SARS-CoV-2, Human Seasonal Influenza H3N2, B. pertussis and M. tuberculosis, and some simulations. They present an index which is calculated per ancestral node of a phylogenetic tree which provides a numeric measure of pathogen population expansion, use the index in an automatic lineage classification scheme, evaluate the determined lineages against the defined lineages (historically / manually / agreed) in each pathogen. They go on to analyse the lineage determining mutations to assess detection of possible phenotypic effects in the phylogeny e.g. do the determined lineages contain founder mutations known to confer different virulence / transmissibility properties.

This is an interesting and useful paper, with the general concept and results in the main body and the more detailed mathematical derivations and simulations in the supplementary. They include the theoretical basis for the index which is good. There is also an implementation of the scheme and worked example in R available on GitHub. I have a few comments relating to the explanation of what has been done and the scientific meaning:

Main text

Lines 37-45: For a general audience I think you need to include another sentence or so on what strain fitness means at population level. You could for example mention more about the SARS-CoV-2 variants, which a general audience would be familiar with.

Lines 54-55: 'not reliably capturing emergent lineages with increased fitness' - yes but you have not really said what fitness is at population level, and anyway fitness is not necessarily the only phenotype of interest when considering public health. Perhaps as mentioned above, if you included a little on the difference between strain fitness (and spreading ability) at a population level vs virulence or pathogenicity at an individual level it would be clearer to a general audience.

Line 81: 'phylogenetically closer' - perhaps explain a little more what you really mean here ? (Although noted it is difficult to not end up writing very technical terms here).

Line 102: 'variant of concern' - do you mean Variant of Concern (VOC) - as in SARS-CoV-2 VOC, or are you referring to your own lineages ? [just change the wording to clarify].

Line 108 and 109: 'population composition' - please clarify - do you mean population composition of lineages, or pathogen effective population size or similar; otherwise it might be confused with host population changes.

Methods

Line 320: 'The alignment was then manually checked' - suggest that you probably checked and adjusted the nucleotide alignment so that it was in-frame ? [i.e. just clarify the wording if required].

Methods - Index definition

Although your Index is a method, it is also a Result and I wonder if there is scope to bring some of this into the Main text ? [although see comments above about reading by a general audience]. Part of the interest of this Index is how the kernel setting bandwidth (b) are selected per pathogen, because your settings (in line 425-429) look important and informative. They are likely influenced by a combination of pathogen specific properties (such as individual waning immunity rate to that pathogen) and population sampling properties, and perhaps these should be highlighted more in the Main text.

Other minor comments on Index definition

Line 360 'phylogenetically closer' - see comments on line 81

Line 366 'isolate to the rest of the population at that time' - i.e. tip to tip in a time window (and not tip to node). ?

Line 377 - perhaps add a short sentence part here to give an indication of what you mean, e.g. 'This timescale ... as well as transmission rate - here we used time scales ranging from months (RNA viruses) to years (bacteria).' ?

Line 374 - so what values of b are actually used for your 4 examples in lines 425-429 ? Maybe include a table in supplementary and reference back to the supplementary here.

Line 431 - perhaps include that the exact timescale does not seem to affect the results for SARS-CoV-2 very much [caption of figure S17].

Supplementary

Please can you re-copy equations 1 and 2 into the supplementary text, so that all of the equations are in the same document [reads more easily].

Maybe include a table of the time scales and kernel b settings for the four real examples (see line 374). [what function of your code allows you to estimate this - is it `index.bandwidth` within `2_1_Index_computation_20231220.R` ?]

Code

Please consider making your calculation code into an R package as well as the GitHub.

Version 1:

Reviewer comments:

Referee #1

(Remarks to the Author)

I thank the authors for their serious attention to my comments.

A simulation study definitely provides an appropriate validation of their method. In my view, the manuscript (and for that matter the response letter) still is not clear enough about what is actual validation and what is simply agreement with previous methods.

The manuscript starts by saying that previous methods use "a priori" lineage definitions. This is not really a correct description since a priori means without looking at the data. It might be that the algorithms used by the approaches they discuss either are (a) not very clear or (b) involve human choice but they are still algorithms of some sort that depend critically on the data.

Instead of "a priori" what the authors mean is that the lineages are "predefined" or better "independently defined" - i.e. defined by another algorithm before selection coefficients are calculated for those lineages. I agree with the authors that this two-step approach may hamper the ability of the methods to identify selection signal but it is not a fundamentally different approach, it relies on the same data. In common with their method, DNA sequence data is used to identify lineages with a selective advantage.

This in clarity of definitions matters because the authors then go on to claim that agreement with previous methods validates their methods. It does not. It simply shows that they are capturing similar signals.

"We found that our framework was able to capture the lineage dynamics of each pathogen considered (Figure 1B-E). Using this framework on SARS-CoV-2 worldwide, we agnostically tracked the changes in population lineage composition, with each main Variant of Concern (VOC) having a clear change in index dynamics (Figure 1B and S2A)."

To be clear, the VOCs were defined based on the same kind of data they are using, there is not a fundamental difference in approach. We do not and cannot know the truth for this real-world data. And that's a good thing for the authors. If we did know the truth, there would be no need for their methods. Indeed the manuscript seems to circle back on itself, first presenting agreement as validation then presenting differences as evidence of different power/applicability of different methods.

Therefore, the simulation study should be presented first and the partial agreement with other analyses be presented as what it is, namely agreement of a formal method with previous less formal (and hence necessarily somewhat haphazard) approaches. It is entirely encouraging when a formal method can capture signals that have previously been identified by

hand as it were, but (to repeat the same point again), humans are computers too.

As it stands, there is a very long paragraph describing the concordance of their results with previous analyses but ... long paragraphs like this are a sign of trouble and I really struggle to know what the take home lesson is. In many cases, they are comparing their selection signals with clades that were simply identified as being phylogenetic branches previously. In general concordance is not that informative about anything as I have tried to explain above, and it would be much better to focus on what are the biological conclusions from their more systematic approach.

The problem of structure is also exemplified by two subheadings that the authors use that also clearly are not... actually distinct, since previous studies are amply referred to in the first subsection.

"Agnostic identification of pathogen lineages"

"Pathogen lineages agnostically identified in the context of previous studies"

This is the main results section of the paper and sorry to say it, it's a mess.

A second area where the manuscript could be more clear on what it is actually they are presenting. It is defined as a "framework", which is somewhat ambiguous but it is given a name "phylowave". Since when were frameworks given names? Phylowave seems instead to be a piece of software (or a pipeline).

The approach would be superior from the point of scientific clarity and repeatability if it were indeed a piece of software (or pipeline) with a user manual, rather than as it is at the moment, a methods section of a paper describing an approach coupled with a set of scripts that allow the user to repeat the specific analyses they present but do not tell the user how to apply the approach to their own data and also as a corollary do not really make it obvious what choices they made in applying the framework to the data that is presented.

I don't think in any case it is really a fully realized "framework" unless it can be applied by others. How do they envisage that this be done? Sorting this out would definitely make the manuscript a more concrete immediate contribution to the field, as well as increasing overall scientific clarity and I do not think it is justifiable to delay this until after publication.

A third area that should be improved is that while I appreciate the simulation study, to describe it as "big" is an exaggeration and in particular they have not tested the method until it breaks, which gives a false impression. It's more in the nature of a minimal simulation study at the moment. All the selection coefficients they have tested are very large and they should do something to ascertain what the parameter space of limited or no statistical power is, since this will be important in defining which signals can be missed by the method. This is key to interpreting the results, for real world data especially for datasets/branches when no signal is found. Also, due to detection bias, I expect the method to systematically overestimate the selection coefficients for mutations that are close to the threshold of detectability, so it's important to know where that threshold is.

The authors claim to present an important new framework to address a biological problem that we can agree is high profile, important and scientifically significant. They should therefore be keen to make it as useable and interpretable as possible, this would surely increase the long-term impact of their interesting work.

Referee #2

(Remarks to the Author)

The authors have done an excellent job with the revisions, I have no further comments.

Referee #3

(Remarks to the Author)

Thanks for considering my points and those of the other reviewers from the previous round of reviews, providing suitable explanations and revising the manuscript. I see that you have now named your framework phylowave, which I think is good and will make it much easier to cite. Also, it is good that you now include some additional simulation results from ReMaster (BEAST2) in the supplementary which I think adds to the paper. I think the revisions and additions have clarified the points well.

Version 2:

Reviewer comments:

Referee #1

(Remarks to the Author)

I thank the reviewers for taking my criticisms onboard wholeheartedly. I think the logical flow of the manuscript has improved substantially and a clearer picture of the relationship to previous methods. The extra simulation results also provide greater understanding of the strengths and limitations of the method. I award myself a reviewer gold star. I have no further suggestions for improvement.

Point-by-point response to reviewers' comments (manuscript 2024-01-00232A)

Referee 1

Comment 1.1: *This paper presents an intriguing looking method for identifying positively-selected lineages within pathogen populations. There are applications to important and topical pathogens. Some nice, although hardly revolutionary stories are told and it does present a nice overview of the evidence of fitness changes in these pathogens.*

Response to comment:

We thank the reviewer for their overall positive feedback. We have responded to the concerns below. In particular, we have now conducted a large number of simulations where we demonstrate we can specifically recover lineages with fitness differences and the associated SNPs.

Comment 1.2: *My main issue with the paper is that it presents a new method but does not test the method in a situation where the ground truth is known, namely simulated data, where a species evolves with occasional fitness-changing mutations on an otherwise neutrally evolving background. If the method is working, then it should be able to identify the lineages with reasonable precision and should include the causal mutation within the set of lineage defining mutations it identifies.*

Response to comment:

We thank the reviewer for their comment. In the previous version of the manuscript, we focused on the mathematical underpinnings of the method and the application to four different pathogens to showcase the approach. However, we agree that a simulation study further tests the robustness of our approach to detect emerging lineages.

In the revised manuscript, we have included the results of a large simulation study. We repeatedly simulated phylogenetic trees where one lineage expands with a known fitness advantage compared to the background population. We tested six different fitness differences, spanning 0.05/unit time to 0.5/time unit. These example values cover a broad range of dynamics ranging from slow to fast lineage replacement, consistent with what we observed in our case study pathogens. We also considered the case of a homogeneous population where no emerging lineage is present. In total, we simulated 700 trees (100 per scenario), each containing 1500 tips. Examples of simulated dynamics are presented in Figure S3 (see below).

In all the scenarios, we find that our method is able to identify the emerging lineages with near-perfect precision (see Figure S4 below, panel A). Our framework is also specific: when no emerging lineage is present, our method does not falsely identify a lineage, in contrast to standard clustering tools, which still define clusters, even in the absence of a lineage with any selective advantage. Further, we quantified the fitness of each lineage by using our logistic model and correctly recovered their fitness (see Figure S4 below, panel B).

Our approach allows us to find the set of mutations that define emerging lineages. To test whether we could recover specifically the causal mutation in each simulation, we simulated sequence alignments on the trees and tried our approach. We found we were able to recover the causal mutation in all the simulations for which we found an emerging lineage (see Figure S4 below, panel C). We note that while we are able to find the causal mutation in each simulation, it is part of a set of mutations associated with the emerging lineage. This is also seen in real data, where multiple mutations can be perfectly associated with a lineage of increased fitness, and therefore it is challenging to find the biologically causal ones. This is not a limitation of our method, but rather due to genetic linkage and population structure, which is an issue for any association method. This can be solved by observing the causal mutation multiple times in different lineages (convergent evolution). This is the case for example in *B. pertussis* where we find substitutions in *sphB1* that are linked to the emergence of two separate lineages, which points towards them being causal mutations.

Specific changes to the manuscript:

In the revised document, we describe the conclusions in the main text in the paragraph "*Agnostic identification of pathogen lineages*": "To assess the robustness of our approach, we repeatedly simulated phylogenetic trees where one lineage expands with a known fitness advantage compared to a background population (Figure S3). In all the scenarios, we found that our method was able to identify the emerging lineage (Figure S4)."

We also provide a full description of this simulation study in the supplementary text, section "*Simulation study to test the ability of our approach to detect emerging lineages*":

"To further test our approach to detect emerging lineages, we conducted a large simulation study. We repeatedly simulated timed phylogenetic trees where one lineage expands with a known fitness advantage compared to the background population. We used the package ReMASTER in BEAST2 to perform the simulations using a coalescent-based approach (Vaughan, 2024). For each scenario, we simulated a tree by separately simulating the background tree and the emerging tree. Each tree was simulated in a coalescent-based fashion, by setting the effective size of each population. The background effective population size was set to 1000 for 40 time units, after which it decreased given a logistic growth model, reflecting the replacement of the background population by the emerging lineage. The emerging effective population size is set to 50 at $T=40$ (equivalent to a proportion of 5% in the population), and increases following a logistic growth model until it reaches 95% of the population, at which point the simulation finishes. The background tree and the emerging tree are then combined to form one unique tree. Trees are subsequently subsampled to include 1500 tips. We also simulated sequence alignments for each simulated tree using the AliSim tool (Ly-Trong et al., 2022) from IQ-TREE (Nguyen et al., 2015). We simulated an alignment of 3000 positions for each tree, with a substitution rate of 0.001 mutations per site per unit of time, an equal proportion of bases and a JC69 substitution model.

We tested six different fitness differences, spanning 0.05/unit time to 0.5/time unit (Table S2). These example values cover a broad range of dynamics ranging from slow to fast lineage replacement, consistent with what we observed in our case study pathogens. We also considered the case of a homogeneous population where no emerging lineage is present. In total, we simulated 700 trees (100 per scenario), each containing 1500 tips. Examples of simulated dynamics are presented in Figure S3.

In all the scenarios, we find that our method is able to identify the emerging lineages with near-perfect precision (Figure S4A). Our framework is also specific: when no emerging lineage is present, our method does not falsely identify a lineage, in contrast to standard clustering tools, which still define clusters, even in the absence of a lineage with any selective advantage. Further, we quantified the fitness of each lineage by using our logistic model and correctly recovered their fitness (Figure S4B). Our approach also allows us to find the set of mutations that define emerging lineages, and particularly the causal mutation (see Figure S4C)."

Figure S3: Example of simulated dynamics for different fitness advantages.

We present examples of simulations in the case of no emerging lineages (A-C) and an emerging lineage with a fitness difference of 0.05 per time unit (D-F), or 0.2 per time unit (G-I). For each condition, we present a simulated tree, the effective population size for each lineage and the index of each node in the simulated tree. Colours denote each population: the background (grey) and the emerging lineage (green). The dashed line represents the time at which the emerging lineage was introduced ($T = 40$).

Figure S4: Results of the simulation study to assess the ability of our approach to detect emerging lineages.

For each scenario, we plot the proportion of times we detect the emerging lineage (A), the estimated fitness difference, using our multinomial logistic model (B) and the proportion of times we detect the causal mutation (C). Dots and bars in A and C denote the median and 95% binomial confidence intervals. The grey triangle in A denote the case where no emerging lineages were simulated. Dots and bars in B denote the median and 95% credible interval of the model posterior.

Comment 1.3: *Being able to recreate previously identified lineages automatically - which is the main validation criterion given by the authors - is not in itself all that impressive. There are other automatic methods out there, as cited by the authors (such as fastBAPS), even if they make slightly different assumptions. And it might be that all of the methods are identifying similar signals, but they are NOT actually signals of selection, but rather simply the same stochastic events. There certainly is not objective evidence that their method is actually better. The robustness to sub-sampling is reassuring but similarly does not actually prove that the method is picking up selection signals as claimed.*

Response to comment:

We agree with the reviewer that a precise comparison of our method to existing ones is important. In the revised manuscript, we test both *treestructure* and *fastbaps* on the simulated datasets described above (see Comment 1.2). Each dataset has one emerging lineage and a background population.

We found that our method was consistently better at recovering the population structure (i.e. one emerging lineage and a background population (see Figure S6, panel A below). More specifically, our method was always able to find the specific emerging lineage in the population. On the contrary, while *treestructure* and *fastbaps* did find the emerging lineage in some cases, they always falsely identified additional lineages within the phylogenies, even when there was no emerging lineage in the simulation (see Figure S6, panel B below). An example of the different methods is presented in Figure S5 below.

Specific changes to the manuscript:

In the revised document, we describe the conclusions in the main text in the paragraph "*Agnostic identification of pathogen lineages*": "Further, we compared our approach to existing methods to agnostically cluster sequences in a tree (*fastbaps*(Tonkin-Hill et al., 2019) and *treestructure*(Volz et al., 2020)) (Figure S5). We found that while *treestructure* and *fastbaps* did find the emerging lineage in some cases, both methods consistently falsely identified additional lineages that did not have any true selective advantage (Figure S6)."

We also provide a full description of how *treestructure* and *fastbaps* were run in the section "*Simulation study to test the ability of our approach to detect emerging lineages*" in the supplementary text:

"To investigate how our framework compares to existing ones, we tested both *fastbaps* (Tonkin-Hill et al., 2019) and *treestructure* (Volz et al., 2020) on the simulated datasets described above. To test *fastbaps*, we simulated sequence alignment for each simulated tree using the *AliSim* tool (Ly-Trong et al., 2022) from *IQ-TREE* (Nguyen et al., 2015). We simulated an alignment of 3000 positions for each tree, with a substitution rate of 0.001 mutations per site per unit of time, an equal proportion of bases and a *JC69* substitution model. We used default parameters to optimise the prior and find the best *baps* partition. To test *treestructure*, we ran the *treestruct* function on each simulated phylogeny with a *minCladeSize* of 30, a significance level of 0.005 and other parameters set to default. For each scenario and for each tree, we then compared the results obtained with each method (Figure S6). An example of the different methods is presented in Figure S5. We found that our method was consistently better at recovering the population structure (i.e. one emerging lineage and a background population."

Figure S6: Comparison of the lineages found by our method, *treestructure* and *fastbaps* on simulated datasets.

For each scenario and each method, we plot the classification agreement (Adjusted Rand Index (Hubert & Arabie, 1985)) (**A**), and the number of lineages detected (**B**). Colours denote the different methods.

Figure S5: Example of classification with our method, treestructure and fastbaps.

We present an example of results on one simulated tree consisting of a background population (grey) and one emerging lineage (green) with a fitness advantage of 0.2 per time unit (**A**). The results with our method are presented in (**B**), treestructure(Volz et al., 2020) in (**C**), and fastbaps(Tonkin-Hill et al., 2019) in (**D**). For each method, the colours indicate the different lineages identified.

Referee 2

Comment 2.1: *The paper of Lefrancoq et al tackles a very important problem, of inferring pathogen lineages that are high fitness and spreading rapidly. The subject is timely in the context of SARS-CoV-2 but not only; and the authors go on to analyze data from four major pathogens, which is laudable. The paper is overall well written and clear, but there are some issues (see below). I am less of an expert on bacterial genomes and thus most of my comments are on the viral data analyzed.*

Response to comment: We thank the reviewer for their comments and enthusiasm about our manuscript.

Comment 2.2: *I have the following comments:*

1. *I liked the idea of agnostic detection of lineages, and I agree with the authors on disadvantages of previous approaches, and especially the need to rely on previous notation of lineages (e.g., PANGO lineages for SARS-CoV-2). However, the use of multinomial logistic regression could also in principle be performed agnostically over all (or some) clades in the tree, similar to what the authors perform (but using a different index). My major comment if so, what would a comparison between these two approaches yield? What advantage (if) is there of the index approach?*

Response to comment:

This is an interesting point. We agree with the reviewer that multinomial logistic models have the advantage of being highly scalable and could in principle be used in a similar way. However, multinomial regression relies on splitting the tree into separate clades. Choosing how to *a priori* cluster sequences into clades is a challenging question, and indeed helped motivate the development of our approach. Existing approaches rely on the structure of a phylogeny to group sequences together, however, this is highly susceptible to uneven sampling over time, and will often result in many singletons. Our approach overcomes this problem by providing a quantitative fitness value (the index) to each tip in the tree, independently of any lineage classification - which is the main advantage of our approach. This allows us to explicitly group tips at each point in time, based on similar index values (i.e., similar fitness dynamics). To demonstrate this point, in the revised manuscript, we repeatedly simulated phylogenetic trees with emerging lineages, we then used standard clade clustering algorithms (*fastbaps* and *treestructure*) to identify clades in the tree, as well as our approach. We found that these standard algorithms could not consistently identify the true emerging lineage. However, our approach was able to recover the true lineage with certainty (see Figure S6 below).

Specific changes to the manuscript:

In the Materials and methods, section "Index definition" we have added the following:

"Our approach provides a quantitative index value to each node (internal and terminal) in the tree, independently of any lineage classification, which is main advantage compared to other methods that rely on lineage classification. This enables to agnostically summarise the changes in population composition in phylogenetic trees at every time point."

Figure S5: Example of classification with our method, treesturcture and fastbaps.

We present an example of results on one simulated tree consisting of a background population (grey) and one emerging lineage (green) with a fitness advantage of 0.2 per time unit (A). The results with our method are presented in (B), treesturcture (Volz et al., 2020) in (C), and fastbaps (Tonkin-Hill et al., 2019) in (D). For each method, the colours indicate the different lineages identified.

Figure S6: Comparison of the lineages found by our method, treesturcture and fastbaps on simulated datasets.

For each scenario and each method, we plot the classification agreement (Adjusted Rand Index (Hubert & Arabie, 1985)) (A), and the number of lineages detected (B). Colours denote the different methods.

Comment 2.3: 2. *The authors highly rely on the phylogeny. At the best of times, it might be better to take into account tree topology uncertainty; and with SARS-CoV-2 (or any pathogen sampled over short time scales), this might become even more profound (see Morel et al. MBE 2021). Could the authors address this shortcoming?*

Response to comment:

We agree with the reviewer that considering uncertainty in the phylogenetic reconstruction is potentially important, although overall tree topology is less relevant as our index uses the genetic

distances between isolates as inputs (rather than the tree shape itself). Nevertheless, there will be uncertainty in genetic distances, due to e.g., uncertainty in nucleotide substitution rates. To test the robustness of the index computation to the exact tree topology, we ran a sensitivity analysis on 3000 trees sampled from the posterior of the BEAST run of *Bordetella pertussis*. We chose *Bordetella pertussis* for this analysis as it is the only pathogen for which we have a posterior distribution of trees. We repeatedly computed the index on each tree sampled from the posterior and computed the average index of each tip. While there is uncertainty in the exact value of the index for each tip, we found that the index dynamics of each lineage remained very consistent across the posterior of trees (Figure S23 below).

Beyond the index computation, we agree that topology uncertainty can have an impact on the exact lineages called by our framework. To improve the accuracy of the detection, we only consider well-supported nodes as potential start of emerging lineages. This includes nodes that have a bootstrap/posterior support of >50% (for *B. pertussis* and *M. tuberculosis*) or nodes that have a least 1 mutation on their directly upstream branch (for SARS-CoV-2 and H3N2).

Specific changes to the manuscript:

In the Material and Methods, in the section 'Index computation on timed-tree with sequences sampled through time' we also include the following:

“To test the robustness of the index computation to the exact tree topology, we ran a sensitivity analysis on 3000 trees sampled from the posterior of the BEAST run of *Bordetella pertussis*. We chose *Bordetella pertussis* for this analysis as it is the only pathogen for which we have a posterior distribution of trees. We repeatedly computed the index on each tree sampled from the posterior and computed the average index of each tip. While there is uncertainty in the exact value of the index for each tip, we found that the index dynamics of each lineage remained very consistent across the posterior of trees (Figure S23).”

Figure S23: Sensitivity analysis: *B. pertussis* Index dynamics over the posterior density of trees

We present the *Index* dynamics computed over 3000 trees from the BEAST posterior of *B. pertussis*. Dots and bars denote the median and 95% credible interval of the *Index* values. We plot only the *Index* of tips as we can summarise their values over the whole posterior.

Comment 2.4: 3. The authors compare their approach to the Obermeyer et al. paper and very briefly write that their “screening recovered the fittest mutations”. More info is necessary here. What proportion? Which mutations are missed out? Which are newly inferred, and can the authors discuss why the discrepancy (as much as possible). Same goes for the Influenza analysis where there is very concrete evidence on antigenic sites and what they mean.

Response to comment:

We apologise that this information was missing. We agree this is important. In the updated version of the manuscript we include more details about which mutations are found by our framework (see below). Overall our findings are consistent with the literature, especially for the well studied pathogens that are SRAS-CoV-2 , H3N2 and *M. tuberculosis*. For *B. pertussis*, a less well-studied bacterium, our framework shed light on two distinct non-synonymous mutations in *sphB1* which are of particular interest as they suggest parallel evolution (Figure S19). *sphB1* encodes a protease which is involved in the extracellular release of the pertussis filamentous haemagglutinin, a *B. pertussis* acellular vaccine antigen and key host-interaction factor. This finding highlights that our framework can identify new genomic changes linked to emerging lineages, and therefore provide testable biological hypotheses into genetic variants in each pathogen that are driving the changes in population fitness of that pathogen.

Specific changes to the manuscript:

We provide more information in the revised manuscript. In the "Lineage-defining mutations." section of the manuscript, we added:

About SARS-CoV-2: "We found that our screening recovered the fittest mutations (all of the 55 fittest mutations, Figure 4I). Among the top 100 fittest mutations, our framework recovered 86% of them. The mutations missed by our method are mainly linked to small subclades of variants, and they seem to have spread in those clades only."

About Influenza H3N2: "We found that indeed, the antigenic sites had the highest proportion of amino acid substitutions compared to the rest of the gene, and that within those, the Koel sites had the highest proportions of substitutions(Koel et al., 2013) (Figure 4J). Among the Koel sites, 86% of positions (N=6, out of 7) were recovered by our framework. The only position missed, 155, is the oldest variable positions, and is not covered by our dataset."

Additionally, in the methods (section "Defining mutations of each lineage"), we provide a full description of the mutations we found:

"For SARS-CoV-2, we matched the amino acid substitution we found to the ones that Obermeyer and colleagues analysed (Obermeyer et al., 2022). The authors analysed 6.4 million genomes up to January 20, 2022 and estimated the fitness effect of 2904 substitutions. Although our global dataset is from an extended period of time (up to 3 April 2023), 83% (N=161) of the lineage-defining mutations were analysed by Obermeyer and colleagues. Our framework was able to recover every single one of the top 55 fittest mutations found by Obermeyer and colleagues. Among the top 100 fittest mutations, our framework recovered 86% of them. The mutations missed by our method are mainly linked to small subclades of variants, and they seem to have spread in those clades only (e.g. in subclades of delta 21I: ORF1a:T3750I and ORF1b:R188Q). One mutation was missed because of a lack of certainty in the ancestral state reconstruction around the root of the Omicron sublineages: S:T376A. From the

mutations that were estimated to have no fitness increase, we found 7 (among 2331 analysed by Obermeyer et al.). These mutations include S:D614G and ORF1b:P314L, two mutations that are linked to an early lineage that eventually got fixed in the whole population. We also found ORF8:G8* and S:G252V, defining our lineage 1 (XBB1*), and ORF1a:L3829F, S:N460K and ORF1b:M1156I, defining our lineage 4 (22E/BQ.1). These mutations were analysed by Obermeyer et al. but were only present in very small frequency in their dataset, as they are mainly linked to the emergence of recent variants, which emerged after their study."

"For H3N2, we computed the proportion of positions that are lineage-defining within each HA polyprotein subunit, and antigenic sites (Koel et al., 2013; Wiley et al., 1981). A position is lineage-defining if it has at least one AA substitution that is lineage-defining. We found that the Koel sites (Koel et al., 2013) had the highest proportion of lineage-defining mutations, with 86% of positions (N=6, out of 7) being recovered by our framework. Specifically, the key positions 156, 159 and 193, defining multiple clades, were recovered. We also recovered positions that defined ancestral lineages, namely positions 145, 158 and 189. Lastly, the position 155 was not picked up by our method as it did not have any variability in our dataset. Indeed, this position was found to be linked to major antigenic changes in the 1960s and 1970s (Koel et al., 2013), which does not overlap with our study period (2005-2023)."

Comment 2.5: 4. *Similar to the above, which known lineages are detected, which are missed? Which non-previously-defined lineages are inferred and why?*

Response to comment:

We agree with the reviewer that a more detailed description of the lineages detected/missed is needed. Overall, we observe a very good concordance between the lineages detected by our framework and the literature for SARS-CoV-2 and *M. tuberculosis*, but the concordance is less good for H3N2 and *B. pertussis*. We would not necessarily expect a high concordance, as we are comparing existing definitions of lineages, often based on unclear, subjective criteria with our new agnostic approach based on fitness differences. Further, while most of the existing definitions are based on mutation definitions on all available data, here we focus on determining which lineages are spreading at a specific scale (i.e. globally for SARS-CoV-2 and H3N2, in France of *B. pertussis* and in Samara, Russia for *M. tuberculosis*). Where there is a low concordance, it suggests that we have identified novel sub-populations with distinct fitness characteristics at those scales.

In the updated manuscript we provide a more precise description of the similarities and differences between the lineages found by our framework and the literature (see below).

Specific changes to the manuscript:

In the section "Pathogen lineages agnostically identified in the context of previous studies." we have included the following text:

For SARS-CoV-2: "The five SARS-CoV-2 VOCs that spread globally were perfectly delineated by our framework (Alpha [B.1.1.7; 20I], Beta [B.1.351; 20H], Gamma [P.1.*; 20J], Delta [B.1.617.2/AY.*;

21A/21J], and Omicron [BA.1.1.529/BA.*; 21K])(Sanyaolu et al., 2021; Viana et al., 2022), and the majority of previously-defined sub-variants were also identified (ARI = 0.80, Figure 2A). We noted that sub-variants that reached a maximum proportion of less than 5% in our global dataset, e.g. Eta/B.1.525, Mu/B.1.621 and EU1, were indistinguishable from others. This highlights the power of our framework in finding lineages that emerge at the geographical scale of the dataset, i.e. globally. Replicating the analysis to SARS-CoV-2 datasets by continent, we re-identify the variants of interest that mainly spread within those continents, e.g. Eta/B.1.525 in Africa, Mu/B.1.621 in the Americas and EU1 in Europe (Figure S8-9)(Hodcroft et al., 2021; Laiton-Donato et al., 2021; Olawoye et al., 2023)."

For H3N2: "For H3N2, while our automatic clades did not match perfectly with the global subclades, the concordance was good (ARI of 0.62). Specifically, a number of global clades were nearly perfectly recovered by our approach (e.g., 3C.3a, 3C.2a3 and 3C.2a1b.1b). Others were mostly well recovered, but with a slight discrepancy in the exact node of emergence (e.g., 3C, 3C.2 and 3C.3). This can be attributed to the index focusing on the signal of lineage expansion rather than a mutation definition. Our agnostic framework mainly differed from the existing classification when considering global clades at a very low frequency in our dataset (for example clades 1*, only 2% of sequences). This further highlights that our framework is focusing on the broad population changes."

For *B. pertussis*: "*B. pertussis*'s population composition is less well-studied. To date, only a few clades have been reported, defined by changes in alleles of the promoter of the pertussis toxin (*ptxP*) and fimbriae 3 gene (*fim3*)(Bart et al., 2014). Our framework was able to find these major clades (ARI = 0.63). We further found three new lineages that emerged. These three lineages have clear distinct index dynamics (Figure 1D, pink, red and purple lineages), but have not been previously identified."

For *M. tuberculosis*: "Further, we recovered most of the known *M. tuberculosis* lineages and sublineages (ARI = 0.92). Specifically, the main global lineages were found(Baker et al., 2004; Casali et al., 2014; Gagneux et al., 2006), with the exception of the distinction between the Central Asian Strain (CAS) and East African Indian (EAI) lineages, which are both present in very small numbers in the dataset and therefore indistinguishable. The SNP-defined sub-lineages were mostly recovered(Homolka et al., 2012), with some discrepancy in lineages such as Harleem, Ural and Latin American-Mediterranean (LAM), which can be attributed to the index focusing on signal of lineage expansion rather than a SNP definition. Therefore our analysis was able to track the expansion of those lineages specifically in Samara, Russia, rather than the global sub-lineages that might have first expanded elsewhere."

Additionally, in the updated version of the manuscript, we now include the data behind the heatmaps in Figure 2 in Tables S2-5. This provides an exhaustive comparison of all the lineages for each pathogen.

Comment 2.6: 5. General comments: while the figures are aesthetically appealing, they are all very hard to follow. First, the figures are tiny as are font sizes. Second, there is a lot of information in each panel that is not always explained.

Response to comment:

We thank the reviewer for their suggestions. We did our best to improve readability in the revised manuscript and have answered the specific points raised below.

Comment 2.7: *Some examples are given but this is rife across all figures:*

a. Fig. 2 the ARI panels on the left, x-axis labels too small to see. The trees aligned against each other are appealing, but the authors need to explicitly write in the text which lineages are concordant, which are not, and why. It would be good to tie this to the ARI displayed

Response to comment:

We apologise for the x-axis labels being too small. In the revised manuscript we have increased the font size of all the text in the Figure 2 (see below). We believe that including both the phylogeny and the heatmap comparing lineage classification of each pathogen is useful for the readers. Nonetheless we appreciate that details might still be hard to read, therefore we have included a new Figure S7 to display the heatmaps in large format (see below).

Additionally, we now describe more precisely in the text which lineages are concordant, which are not, and why (see comment 2.5 above). we now include the data behind the heatmaps in Figure 2 and S7 in Tables S2-5. This provides an exhaustive comparison of all the lineages for each pathogen.

Figure 2: Comparison of the identified lineages to the known population composition.

For each pathogen, we present a heatmap comparing the known population structure (x-axis) to the automatic clades found by our framework (y-axis). Darker colours represent more agreement between both classifications. Contingency tables are presented in Tables S2-5. The heatmaps are presented in large in Figure S7. We also compare the timed-resolved phylogenetic trees coloured by respective lineage classifications: automatic clades on the left, and previously identified lineages on the right. The colours of the automatic clades are the same as in Figure 1. For *M. tuberculosis*, LAM denotes the Latin American-Mediterranean lineage, E-A the Euro-American lineage, EAI the East African Indian lineage and CAS the Central Asian Strain lineage.

Figure S7: Heatmaps comparing the identified lineages to the known population composition. For each pathogen, we present a heatmap comparing the known population structure (x-axis) to the automatic clades found by our framework (y-axis). Darker colours represent more agreement between both classifications. Contingency tables are presented in Tables S2-5.

Comment 2.8: *b. I have no idea what figure 4I is presenting*

Response to comment:

We apologise for the confusion. In Figure 4I we compare the substitutions analysed by Obermeyer and colleagues to the mutations found to be lineage-defining in our study. For each amino acid change, the figure presents the estimated increase in fitness $\Delta \log R$ (y axis), as a function of its rank, inferred by statistical significance (x axis). We colour in red the mutations found by our method. This figure depicts that our method is able to recover the fittest mutations.

Specific changes to the manuscript:

In the revised manuscript, we have added into the legend of Figure 4I (see below comment 2.9): "For SARS-CoV-2, we compare the substitutions analysed by Obermeyer and colleagues (Obermeyer et al., 2022) and the mutations found to be lineage-defining in our study. For each amino acid change, the figure presents the estimated increase of fitness $\Delta \log R$ (y axis), as a function of its rank, inferred by statistical significance (x axis). Mutations in red are found by our method."

Comment 2.9: *c. Figure 4A-D shows lineage, but this is meaningless unless some biological context is given for all lineages and discussed (comment 4).*

Response to comment:

We chose to present the lineage trees in Figure 4A-D as a summary of the lineages found in our study. In the revised manuscript we have updated them with more labels to improve readability and biological relevance (see below). Further, as discussed above, the lineages highlighted in the figure are now discussed in the text.

Figure 4: Lineage-defining genetic mutations.

For each pathogen, we present a summary of the genetic evolution of the lineages. (A-D) For each pathogen, we present the lineage trees representing the genealogical relationship between them. Key clades are highlighted. Colours indicate groups. (E-H) Lineage-defining mutations along the genome of each pathogen considered. For SARS-CoV-2 (E) and H3N2 (F) viruses we plot the density of lineage-defining mutations along the full genome (SARS-CoV-2) or HA polyprotein (H3N2). Colours indicate the main ORFs. For *B. pertussis* (G) and *M. tuberculosis* (H) we plot for each mutation the maximum proportion of that mutation that is present in any group (*B. pertussis*) or in groups 1 and 2 (*M. tuberculosis*). The dashed lines represent the 0.8 cutoff. The lists of mutations identified can be found in Data Files S5-8. (I-L) Functional relevance of the mutations identified. (I) For SARS-CoV-2, we compare the substitutions analysed by Obermeyer and colleagues (Obermeyer et al., 2022) and the mutations found to be lineage-defining in our study. For each amino acid change, the figure presents the estimated increased of fitness $\Delta \log R$ (y axis), as a function of its rank, inferred by statistical significance (x axis). Mutations in red are found by our method. (J) For H3N2, we plot the proportion of positions that are lineage-defining within each HA polyprotein subunit, and antigenic sites (Koel et al., 2013; Wiley et al.,

1981)(insert). (K) For *B. pertussis*, we plot the proportion of mutations that are lineage-defining within each functional category (Parkhill et al., 2003) (L) Same as K, for *M. tuberculosis* (Chitale et al., 2022). The lists of lineage-defining mutations for each pathogen can be found in DataFiles S5-8.

Comment 2.10: *d. in the ARI plots, in some cases it seems very low concordance, can the authors comment on this.*

Response to comment:

We agree that the concordance of SARS-CoV-2 and *M. tuberculosis* are very good, and those of Influenza H3N2 and *B. pertussis* are less good. However, we stress that we would not necessarily expect a high concordance, as we are comparing existing definitions of lineages, often based on unclear, subjective criteria with our new agnostic approach based on fitness differences. Where there is a low concordance, it suggests that we have identified novel sub-populations with distinct fitness characteristics.

In the case of *B. pertussis*, it suggests that the genotypes defined based on the alleles of *ptxP* and *fim3* do not capture the entire fitness dynamics of the bacterium in France. Indeed, the major splits between the genotypes are found by our framework, but we also identify novel sub-populations.

In the case of H3N2, it suggests that the existing clade nomenclature approach is not capturing the true underlying heterogeneity in fitness between lineages. The high concordance for SARS-CoV-2 is not too surprising as there were clear sequential selective sweeps in the population, facilitating the identification of lineages with clear fitness differences.

Specific changes to the manuscript:

We now include in the 'Pathogen lineages agnostically identified in the context of previous studies' section: "We found the level of concordance differed by pathogen (mean ARI of 0.75 across pathogens, min 0.62, max 0.94), suggesting variability in the ability of existing approaches to appropriately capture the heterogeneity in lineages with distinct fitness dynamics."

Comment 2.11: *6. If I understand correctly, Figure 3E-H shows a pattern of initial negative fitness that becomes positive fitness for almost all lineages found. Is this a biological feature (changing antigenic landscape) or is this an issue with sample size increasing? Why is this different across the four pathogens (e.g., for influenza seems much less prominent)?*

Response to comment:

We thank the reviewer for this comment, which enabled us to spot a colour key issue. The colour key was slightly skewed, causing positive values close to 0 to be in blue. The fitness of lineage at the time of emergence is always positive, as it is starting to replace the population.

We have now updated Figure 3 with the correct colour key:

Figure 3: Estimation of the fitness of each lineage.

(A-D) Model fits per pathogen. For each pathogen, we present the fits for the five most prevalent groups. The fits for all groups are presented in Figure S11-14. Coloured dots represent data, bars denote 95% confidence intervals. Coloured lines and shaded areas represent the median and 95% credible interval of the posterior. (E-H) Relative fitness of each group, over time. Estimates for all groups are presented in Figure S15. Crosses indicate the group's MRCA. Open circles indicate the last isolate from each group, in our datasets.

Comment 2.12: Minor comments:

1. Line 63, the authors argue that some methods are susceptible to sampling biases, this is very true for the author's approach as well.

Response to comment:

We agree with the reviewer that all phylogenetic methods, including ours, are subject to sampling bias. However, when it comes to the most common type of sampling bias, i.e. varying sampling intensity over time, we have demonstrated that our method performs very well (see Figure 5). Nonetheless, to acknowledge that our method is still subject to sampling biases, especially those that could affect the sampled diversity, we now include a sentence in the 'Tracking lineages in real-time' section: "As other phylodynamic methods, we note that our framework is sensitive to biases in the source of sequence, particularly where the sequenced pathogens are not representative of the diversity of the underlying population."

Comment 2.13: 2. How much does the method rely on the sampling strategies employed by NextStrain. If the authors were to perform sampling on their own, how would their method fair?

Response to comment:

Nextstrain subsamples publically available genome data to a reasonable degree, they provision multiple views to focus subsampling with different geographic regions and different time periods. We used their sampling strategy for simplicity. If we were to perform sampling ourselves we would have used a similar strategy (i.e. selecting ~3000 sequences over the time period of 2020 to mid-2023, with a balanced representation of locations). Therefore, we expect that if we were to perform sampling of our own it would yield near-identical results.

Supporting this point, in our study we performed multiple analyses on different SARS-CoV-2 datasets that yielded similar results. First, we analysed continent-specific SARS-CoV-2 datasets (Figure S8-9), and found that our framework was able to re-identify the variants of interest that mainly spread within those continents. As each dataset and sub-sampling was independent this highlights that our results are robust to the specific sampling chosen. Further, in a sensitivity analysis, we performed our own sub-sampling on the NextStrain data and showed that our framework was able to detect virtually all the lineages (Figure 5), even in biased conditions.

Comment 2.14: 3. While I understood the logic behind the index via figure 1, I had a hard time understanding the index defined in eq.1. Why is the summation over d rather than over nodes? Why are we summing over a distribution over and over?

Response to comment:

We apologise for the confusion, this is indeed a key point. As the reviewer pointed out, the index can be computed by summing over all nodes, as shown by this equation from the "Index computation on timed-tree with sequences sampled through time" section in the methods:

$$Index(i) = \sum_{j \in nodes} I(t_j > t_i - t_{wind} \& t_j < t_i + t_{wind}) d(i,j) b^{d(i,j)} \quad [Eq. 3]$$

Where $nodes$ is the set of all nodes in the tree, I is an indicator function, and $d(i, n)$ is the distance between nodes i and j .

This Equation 3 is mathematically equivalent to the Equation 1 presented in the manuscript:

$$Index(i) = \sum_{d=0}^{\infty} D_i(d, t) \cdot b^d \quad [Eq. 1]$$

Indeed, instead of computing the index by summing on each node in the population, we first compute the distance distribution from node i to the rest of the population, and sum over this distance distribution.

The reason why we chose to sum over the distance distribution rather than nodes is that we can mathematically compute those theoretical pairwise distance distributions. Therefore, this formulation of the index definition enables us to write an expectation of the index dynamics over time for different populations.

Specific changes to the manuscript:

In the revised manuscript we now label Equation 3 properly (it was previously unlabeled), and we added after the description of Equation 1: "In practice, to compute the index of each node in a phylogenetic tree, we sum over the distance to all nodes in the population, rather than the distance distribution (see section "Index computation on timed-tree with sequences sampled through time")."

Comment 2.15: 4. *Does evolutionary time (below figure 1) correspond to branch lengths?*

Response to comment: Yes, we have updated the line to make it clearer: "With $D_i(d, t)$ the distance distribution - in number of mutations or evolutionary time (branch length) - [...]"

Comment 2.16: 5. *How can the bandwidth b equal zero?*

Response to comment: Thank you for pointing out this typo, the branch length cannot be zero. We have updated the formal definition of b to: $b \in]0,1[$

Comment 2.17: 6. *Figure S18 – I can't really see an "elbow" – which means the choice of the number of lineages is rather arbitrary.*

Response to comment: In principle, for each pathogen, we choose the number of groups at which the deviance explained by the model does not increase substantially any more. We agree with the reviewer that the 'elbow' is hard to see in many cases, which is why we used the log scale in Figure S24 (number in the revised manuscript, corresponding to Figure S18 in the previous version).

Theoretically, an elbow is expected. In the revised manuscript, we include a new Figure S25 from our simulation study. Similar to Figure S24, we plot the non-explained deviance as a function of the number of groups for each simulation of each scenario. A clear elbow at $N_{groups} = 2$ is seen for the scenarios where there is an emerging lineage. In the scenario in which there is no lineage, no elbow is found.

On real data, as for any clustering algorithm, choosing the best number of lineages that describe the index dynamics is a challenging question. Ultimately the user has to make a choice, where increasing the number of distinct lineages will progressively identify lineages with increasingly reduced fitness differences. We believe that these elbow plots are a useful guide possible to choose the appropriate number of lineages within a timed phylogeny. We have included this guidance in the revised manuscript.

Figure S25: Simulation study: non-explained deviance as a function of the number of groups in the lineage detection algorithm.

We simulated trees with or without an emerging lineage - from left to right: no lineage, fitness difference of 0.05/time unit, and fitness difference of 0.2/time unit. We present here the proportion of non-explained deviance by the models with different numbers of groups. Each simulation is plotted as a line (100 lines per scenario). The colour of the line indicates if a lineage was detected (grey: no lineage, red: one lineage detected). Lineages are detected if the non-explained deviance continues to decrease. Dashed lines denote the number of groups chosen.

Specific changes to the manuscript:

We now include in the 'Agnostic detection of lineages' section in the Methods: 'In simulations, we demonstrate a clear elbow that precisely identifies the optimal number of discrete lineages in the dataset (Figure S25). However, not all pathogens in real-world datasets will give clear 'elbows'. This may be due to insufficient sampling intensity and the presence of lineages with only very small differences in fitness. In practice, increasing the number of distinct lineages will progressively lead to the identification of lineages with increasingly reduced fitness differences, with increasing risks of falsely dividing lineages into subpopulations where no true difference exists.'

Comment 2.18: 7. *Why are lineage defining mutation not inferred directly from the tree, as the mutations occurring on the lineage leading to the ancestral node?*

Response to comment: Lineage-defining mutations are not inferred directly from the tree as there might be some uncertainty in the exact timing of the emerging lineage, and the tree itself will depend on sequencing intensity over time. This means that relying on sequence reconstruction at a specific node in the tree is suboptimal. In particular, a lineage can take some time to start growing and be detectable. We therefore use a comparison of sequences between different lineages to identify the specific mutations that are different.

Specific changes to the manuscript:

We now include: "We did not infer lineage-defining mutations directly from the tree as there might be some uncertainty in the exact timing of the emerging lineage, and the tree itself will depend on sequencing intensity over time. In particular, a lineage can take some time to start growing and be detectable. We instead used a comparison of sequences between different lineages to identify the specific mutations that are different. "

Referee 3

Comment 3.1: *The paper by Lefrancq et al is about analysing phylogenetic trees created from DNA (or RNA) sequences of viruses and bacteria, which have been collected from a host population (e.g. humans) over a time period of typically some years. They use four real exemplar situations - SARS-CoV-2, Human Seasonal Influenza H3N2, B. pertussis and M. tuberculosis, and some simulations. They present an index which is calculated per ancestral node of a phylogenetic tree which provides a numeric measure of pathogen population expansion, use the index in an automatic lineage classification scheme, evaluate the determined lineages against the defined lineages (historically / manually / agreed) in each pathogen. They go on to analyse the lineage determining mutations to assess detection of possible phenotypic effects in the phylogeny e.g. do the determined lineages contain founder mutations known to confer different virulence / transmissibility properties.*

This is an interesting and useful paper, with the general concept and results in the main body and the more detailed mathematical derivations and simulations in the supplementary. They include the theoretical basis for the index which is good. There is also an implementation of the scheme and worked example in R available on GitHub. I have a few comments relating to the explanation of what has been done and the scientific meaning:

Response to comment: We thank the reviewer for their thorough review. We have responded to their comments below.

Comment 3.2: *Main text*

Lines 37-45: For a general audience I think you need to include another sentence or so on what strain fitness means at population level. You could for example mention more about the SARS-CoV-2 variants, which a general audience would be familiar with.

Response to comment: We agree with the reviewer that including additional detail on what strain fitness means at the population level is important.

We have edited the first paragraph of our manuscript to define the meaning of fitness at the population level :

"For most pathogens, there are constantly changing patterns of strain composition. Pressures to evade host immunity, environmental shifts or changing abilities to infect and disseminate in hosts result in the emergence of some lineages with increased fitness and the extinction of others. These dynamic patterns of genetic diversity are a fundamental aspect of disease ecology. They also have potentially critical public health implications, including signifying immune or vaccine escape or improved transmissibility. It has, however, been difficult to identify and quantify lineages with differential levels of fitness, i.e. different ability to spread in the population, especially outside highly genetically sampled pathogen systems such as SARS-CoV-2 or influenza(Lefrancq et al., 2022; Luksza & Lässig, 2014; Meijers et al., 2023). Identifying lineages with improved fitness at the population level would

allow focused public health response, through e.g., targeted vaccination, as well as provide key insights into the underlying ecology of disease systems."

Comment 3.3: *Lines 54-55: 'not reliably capturing emergent lineages with increased fitness' - yes but you have not really said what fitness is at population level, and anyway fitness is not necessarily the only phenotype of interest when considering public health. Perhaps as mentioned above, if you included a little on the difference between strain fitness (and spreading ability) at a population level vs virulence or pathogenicity at an individual level it would be clearer to a general audience.*

Response to comment: We agree with the reviewer, as detailed above, we have edited the first paragraph of our manuscript:

"For most pathogens, there are constantly changing patterns of strain composition. Pressures to evade host immunity, environmental shifts or changing abilities to infect and disseminate in hosts result in the emergence of some lineages with increased fitness and the extinction of others. These dynamic patterns of genetic diversity are a fundamental aspect of disease ecology. They also have potentially critical public health implications, including signifying immune or vaccine escape or improved transmissibility. It has, however, been difficult to identify and quantify lineages with differential levels of fitness, i.e. different ability to spread in the population, especially outside highly genetically sampled pathogen systems such as SARS-CoV-2 or influenza (Lefrancq et al., 2022; Luksza & Lässig, 2014; Meijers et al., 2023). Identifying lineages with improved fitness at the population level would allow focused public health response, through e.g., targeted vaccination, as well as provide key insights into the underlying ecology of disease systems."

Comment 3.4: *Line 81: 'phylogenetically closer' - perhaps explain a little more what you really mean here? (Although noted it is difficult to not end up writing very technical terms here).*

Response to comment: We agree the term "phylogenetically" is quite specific, we have modified the sentence as follows:

"This measure is based on the expectation that nodes sampled from an emerging fitter lineage will be phylogenetically closer than the rest of the population at that time, as they will all share the same recent ancestor."

Comment 3.5: *Line 102: 'variant of concern' - do you mean Variant of Concern (VOC) - as in SARS-CoV-2 VOC, or are you referring to your own lineages? [just change the wording to clarify].*

Response to comment: We apologise for the confusion, we mean the Variant of Concern (VOC), as shown in Figure S2A. The sentence now reads "*with each main Variant of Concern (VOC) having a clear change in index dynamics (Figure 1B and S2A).*"

Comment 3.6: *Line 108 and 109: 'population composition' - please clarify - do you mean population composition of lineages, or pathogen effective population size or similar; otherwise it might be confused with host population changes.*

Response to comment: We mean the population composition in terms of lineages. We have changed "population composition" to "population lineage composition".

Comment 3.7: *Methods*

Line 320: 'The alignment was then manually checked' - suggest that you probably checked and adjusted the nucleotide alignment so that it was in-frame ? [i.e. just clarify the wording if required].

Response to comment: Indeed, we checked that the alignment was in-frame and had minimal gaps. We have modified the sentence as follows: "We then manually checked that the alignment did not have any frameshifts and had minimal gaps."

Comment 3.8: *Methods - Index definition*

Although your Index is a method, it is also a Result and I wonder if there is scope to bring some of this into the Main text ? [although see comments above about reading by a general audience]. Part of the interest of this Index is how the kernel setting bandwidth (b) are selected per pathogen, because your settings (in line 425-429) look important and informative. They are likely influenced by a combination of pathogen specific properties (such as individual waning immunity rate to that pathogen) and population sampling properties, and perhaps these should be highlighted more in the Main text.

Response to comment: We thank the reviewer for this comment. We agree that our framework is a result. However, we believe that adding some of the methods in the main text might result in the manuscript being hard to read for non-specialist readers. We describe in the paragraph "Tracking population composition in timed phylogenetic trees" how our framework works in plain words.

As the reviewer points out, selecting the timescale of the kernel is important, and will depend on both the transmission rate of the pathogen in the population and the strength of the molecular signal. This is highlighted in the "Tracking population composition in timed phylogenetic trees." paragraph in the manuscript: "The timescale is tailored to the specific pathogen studied and its choice will depend on the molecular signal, as well as the transmission rate."

Ultimately, as the reviewer points out, the timescale does modify the perceived index dynamics, but our framework results remain unchanged (see below, Figure S22).

Figure S22: Robustness of the framework to the choice of timescale

We show our framework is robust to the choice of the timescale (governing the weight distribution used in the index computation). **(A-B)** Index dynamics computed on the global SARS-CoV-2 tree, with either a timescale of 0.075 (A) or 0.3 (B). The timescale used in the main analysis is 0.15. A smaller timescale focused more on recent population dynamics, a larger timescale focused more on the past evolution. Colours represent the lineages identified with our algorithm on those dynamics. **(C-D)** We compare the lineages identified with those timescales (y-axis) to the lineages presented throughout this study, with a timescale of 0.15 (x-axis). Darker colours represent more agreement between both namings. Overall, we find minimal differences in the lineages detected. Our results for SARS-CoV-2 are robust to the exact chosen timescale.

Comment 3.9: Other minor comments on Index definition

Line 360 'phylogenetically closer' - see comments on line 81

Response to comment: Similarly as the comment 3.4, we have updated this sentence as follows:

"This measure is based on the expectation that sequences sampled from an emerging, fitter, lineage will be phylogenetically closer than the rest of the population at that time, as they will all share the same recent ancestor."

Comment 3.10: Line 366 - 'isolate to the rest of the population at that time' - i.e. tip to tip in a time window (and not tip to node). ?

Response to comment: When computing the distance distributions we consider that internal nodes are representative of members of the past population and therefore take them into account in the computation. 'isolate to the rest of the population at that time' therefore, means tip to any node in the population (internal and external).

We have updated this sentence as follows (including the comment from reviewer 2 and comments below): "With $D_i(d, t)$ the distance distribution - in number of mutations or evolutionary time (branch length) - from the isolate i to the rest of the population (internal and terminal nodes) at that time t (Figure 1) and b^d , the kernel setting the weight of each distance d . b is the bandwidth, $b \in]0,1[$, which is a parameter to set, linked to the timescale (Table S1)."

Comment 3.11: Line 377 - perhaps add a short sentence part here to give an indication of what you mean, e.g. 'This timescale ... as well as transmission rate - here we used time scales ranging from months (RNA viruses) to years (bacteria).' ?

Response to comment: We thank the reviewer for the suggestion, we have added a sentence at the end of the paragraph: "Here we used timescales ranging from 1.8 months (SARS-CoV-2, RNA virus) to 30 years (*Mycobacterium tuberculosis*, bacteria) (Table S1)."

Comment 3.12: Line 374 - so what values of b are actually used for your 4 examples in lines 425-429 ? Maybe include a table in supplementary and reference back to the supplementary here.

Response to comment: Indeed this information was not present in our manuscript, we thank the reviewer for pointing this out. We now provide the values of the time scales and associated kernel bandwidth parameters b in Table S1. We have referenced this table throughout the manuscript.

Pathogen	SARS-CoV-2	Influenza H3N2	B. pertussis	M. tuberculosis
Genome length	29903	1701	4086189	4411532
Substitution rate (substitution per site per year)	$8.1 \cdot 10^{-4}$	$3.82 \cdot 10^{-3}$	$2.5 \cdot 10^{-7}$	$4.6 \cdot 10^{-8}$
Time scale (years)	0.15	0.4	2	30
Kernel bandwidth b	0.91	0.87	0.84	0.94

Table S1: Genome lengths, substitution rates, timescales and bandwidths used in this study.

We present the list of the different parameters that we use to compute the index. The function "*index.bandwidth()*" in the GitHub library allows to compute the kernel bandwidth given a genome length, a substitution rate and a timescale. The kernel bandwidth is dimensionless.

Comment 3.13: Line 431 - perhaps include that the exact timescale does not seem to affect the results for SARS-CoV-2 very much [caption of figure S17].

Response to comment: We have added "Our results for SARS-CoV-2 are robust to the exact chosen timescale." at the end of the legend.

Comment 3.14: Supplementary

Please can you re-copy equations 1 and 2 into the supplementary text, so that all of the equations are in the same document [reads more easily].

Response to comment: We thank the reviewer for the suggestion, we have added equations 1 and 2 to the supplementary text. This indeed improves readability.

Comment 3.15: Maybe include a table of the time scales and kernel b settings for the four real examples (see line 374). [what function of your code allows you to estimate this - is it `index.bandwidth` within `2_1_Index_computation_20231220.R` ?]

Response to comment: We thank the reviewer for the suggestion. We have included in Table S1 the time scales, genome parameters and corresponding kernel bandwidths for each pathogen. Further, we have referenced the "`index.bandwidth()`" function in both the methods and the legend of the table to allow the readers to use the code if needed.

Pathogen	SARS-CoV-2	Influenza H3N2	B. pertussis	M. tuberculosis
Genome length	29903	1701	4086189	4411532
Substitution rate (substitution per site per year)	$8.1 \cdot 10^{-4}$	$3.82 \cdot 10^{-3}$	$2.5 \cdot 10^{-7}$	$4.6 \cdot 10^{-8}$
Time scale (years)	0.15	0.4	2	30
Kernel bandwidth b	0.91	0.87	0.84	0.94

Table S1: Genome lengths, substitution rates, timescales and bandwidths used in this study.

We present the list of the different parameters that we use to compute the index. The function "`index.bandwidth()`" in the GitHub library allows to compute the kernel bandwidth given a genome length, a substitution rate and a timescale. The kernel bandwidth is dimensionless.

Comment 3.16: Code.

Please consider making your calculation code into an R package as well as the GitHub.

Response to comment: We are working on making the code available as a package.

Referee 3 (Remarks on code availability)

Comment 3.17: *The code for performing and plotting the index calculations and doing automatic lineage detection is available on the author's site on GitHub. The code is written in R and the SARS-CoV-2 example worked is provided using R markdown (open in Rstudio), runs well and generates the figures [I found slight issues with using ggtree, but that is a dependency problem not a problem with this code]. There are also separate and suitably documented functions for the individual calculations.*

Additionally, I would encourage the authors to make the calculation functions available as an R package (or part of R package) ideally on Cran, and also provide an example of useage with some TreeSim simulated trees for the R package; that way it is all self contained and you dont need to provide real trees just to show code operation only (but including your real trees in the package would be useful).

Response to comment: We thank the reviewer for taking the time to review the code and are pleased that everything ran as planned. We are working on making the code available as a package.

References

- Baker, L., Brown, T., Maiden, M. C., & Drobniowski, F. (2004). Silent nucleotide polymorphisms and a phylogeny for *Mycobacterium tuberculosis*. *Emerging Infectious Diseases*, *10*(9), 1568–1577.
- Bart, M. J., Harris, S. R., Advani, A., Arakawa, Y., Bottero, D., Bouchez, V., Cassidy, P. K., Chiang, C.-S., Dalby, T., Fry, N. K., Gaillard, M. E., van Gent, M., Guiso, N., Hallander, H. O., Harvill, E. T., He, Q., van der Heide, H. G. J., Heuvelman, K., Hozbor, D. F., ... Mooi, F. R. (2014). Global population structure and evolution of *Bordetella pertussis* and their relationship with vaccination. *mBio*, *5*(2), e01074.
- Casali, N., Nikolayevskyy, V., Balabanova, Y., Harris, S. R., Ignatyeva, O., Kontsevaya, I., Corander, J., Bryant, J., Parkhill, J., Nejentsev, S., Horstmann, R. D., Brown, T., & Drobniowski, F. (2014). Evolution and transmission of drug-resistant tuberculosis in a Russian population. *Nature Genetics*, *46*(3), 279–286.
- Chitale, P., Lemenze, A. D., Fogarty, E. C., Shah, A., Grady, C., Odom-Mabey, A. R., Johnson, W. E., Yang, J. H., Eren, A. M., Brosch, R., Kumar, P., & Alland, D. (2022). A comprehensive update to the *Mycobacterium tuberculosis* H37Rv reference genome. *Nature Communications*, *13*(1), 7068.
- Gagneux, S., DeRiemer, K., Van, T., Kato-Maeda, M., de Jong, B. C., Narayanan, S., Nicol, M., Niemann, S., Kremer, K., Gutierrez, M. C., Hilty, M., Hopewell, P. C., & Small, P. M. (2006). Variable host-pathogen compatibility in *Mycobacterium tuberculosis*. *Proceedings of the National Academy of Sciences of the United States of America*, *103*(8), 2869–2873.
- Hodcroft, E. B., Zuber, M., Nadeau, S., Vaughan, T. G., Crawford, K. H. D., Althaus, C. L., Reichmuth, M. L., Bowen, J. E., Walls, A. C., Corti, D., Bloom, J. D., Velesler, D., Mateo, D., Hernando, A., Comas, I., González-Candelas, F., SeqCOVID-SPAIN consortium, Stadler, T., & Neher, R. A. (2021). Spread of a SARS-CoV-2 variant through Europe in the summer of 2020. *Nature*, *595*(7869), 707–712.
- Homolka, S., Projahn, M., Feuerriegel, S., Ubben, T., Diel, R., Nübel, U., & Niemann, S. (2012). High resolution discrimination of clinical *Mycobacterium tuberculosis* complex strains based on single nucleotide polymorphisms. *PLoS One*, *7*(7), e39855.
- Hubert, L., & Arabie, P. (1985). Comparing partitions. *Journal of Classification*, *2*(1), 193–218.
- Koel, B. F., Burke, D. F., Bestebroer, T. M., van der Vliet, S., Zondag, G. C. M., Vervaeke, G., Skepner, E., Lewis, N. S., Spronken, M. I. J., Russell, C. A., Eropkin, M. Y., Hurt, A. C., Barr, I. G., de Jong, J. C., Rimmelzwaan, G. F., Osterhaus, A. D. M. E., Fouchier, R. A. M., & Smith, D. J. (2013). Substitutions near the receptor binding site determine major antigenic change during influenza virus evolution. *Science*, *342*(6161), 976–979.
- Laiton-Donato, K., Franco-Muñoz, C., Álvarez-Díaz, D. A., Ruiz-Moreno, H. A., Usme-Ciro, J. A., Prada, D. A., Reales-González, J., Corchuelo, S., Herrera-Sepúlveda, M. T., Naizaque, J., Santamaría, G., Rivera, J., Rojas, P., Ortiz, J. H., Cardona, A., Malo, D., Prieto-Alvarado, F., Gómez, F. R., Wiesner, M., ... Mercado-Reyes, M. (2021). Characterization of the emerging B.1.621 variant of interest of SARS-CoV-2. *Infection, Genetics and Evolution: Journal of Molecular Epidemiology and Evolutionary Genetics in Infectious Diseases*, *95*, 105038.
- Lefrancq, N., Bouchez, V., Fernandes, N., Barkoff, A.-M., Bosch, T., Dalby, T., Åkerlund, T., Darenberg, J., Fabianova, K., Vestrheim, D. F., Fry, N. K., González-López, J. J., Gullsby, K., Habington, A., He, Q., Litt, D., Martini, H., Piérard, D., Stefanelli, P., ... Brisse, S. (2022). Global spatial dynamics and vaccine-induced fitness changes of *Bordetella pertussis*. *Science Translational Medicine*, *14*(642), eabn3253.
- Luksza, M., & Lässig, M. (2014). A predictive fitness model for influenza. *Nature*, *507*(7490), 57–61.
- Ly-Trong, N., Naser-Khdour, S., Lanfear, R., & Minh, B. Q. (2022). AliSim: A Fast and Versatile Phylogenetic Sequence Simulator for the Genomic Era. *Molecular Biology and Evolution*, *39*(5). <https://doi.org/10.1093/molbev/msac092>
- Meijers, M., Ruchnewitz, D., Eberhardt, J., Łuksza, M., & Lässig, M. (2023). Population immunity predicts evolutionary trajectories of SARS-CoV-2. *Cell*. <https://doi.org/10.1016/j.cell.2023.09.022>
- Nguyen, L.-T., Schmidt, H. A., von Haeseler, A., & Minh, B. Q. (2015). IQ-TREE: a fast and effective stochastic algorithm for estimating maximum-likelihood phylogenies. *Molecular Biology and Evolution*, *32*(1), 268–274.
- Obermeyer, F., Jankowiak, M., Barkas, N., Schaffner, S. F., Pyle, J. D., Yurkovetskiy, L., Bosso, M., Park, D. J., Babadi, M., Maclinnis, B. L., Luban, J., Sabeti, P. C., & Lemieux, J. E. (2022). Analysis of 6.4 million SARS-CoV-2 genomes identifies mutations associated with fitness. *Science*, *376*(6599), 1327–1332.
- Olawoye, I. B., Oluniyi, P. E., Oguzie, J. U., Uwanibe, J. N., Kayode, T. A., Olumade, T. J., Ajogbasile, F. V., Parker, E., Eromon, P. E., Abechi, P., Sobajo, T. A., Ugwu, C. A., George, U. E., Ayoade, F., Akano, K., Oyejide, N. E., Nosamiefan, I., Fred-Akintunwa, I., Adedotun-Sulaiman, K., ... Happi, C. T. (2023). Emergence and

- spread of two SARS-CoV-2 variants of interest in Nigeria. *Nature Communications*, 14(1), 811.
- Parkhill, J., Sebahia, M., Preston, A., Murphy, L. D., Thomson, N., Harris, D. E., Holden, M. T. G., Churcher, C. M., Bentley, S. D., Mungall, K. L., Cerdeño-Tárraga, A. M., Temple, L., James, K., Harris, B., Quail, M. A., Achtman, M., Atkin, R., Baker, S., Basham, D., ... Maskell, D. J. (2003). Comparative analysis of the genome sequences of *Bordetella pertussis*, *Bordetella parapertussis* and *Bordetella bronchiseptica*. *Nature Genetics*, 35(1), 32–40.
- Sanyaolu, A., Okorie, C., Marinkovic, A., Haider, N., Abbasi, A. F., Jaferi, U., Prakash, S., & Balendra, V. (2021). The emerging SARS-CoV-2 variants of concern. *Therapeutic Advances in Infectious Disease*, 8, 20499361211024372.
- Tonkin-Hill, G., Lees, J. A., Bentley, S. D., Frost, S. D. W., & Corander, J. (2019). Fast hierarchical Bayesian analysis of population structure. *Nucleic Acids Research*, 47(11), 5539–5549.
- Vaughan, T. G. (2024). ReMASTER: improved phylodynamic simulation for BEAST 2.7. *Bioinformatics*, 40(1). <https://doi.org/10.1093/bioinformatics/btae015>
- Viana, R., Moyo, S., Amoako, D. G., Tegally, H., Scheepers, C., Althaus, C. L., Anyaneji, U. J., Bester, P. A., Boni, M. F., Chand, M., Choga, W. T., Colquhoun, R., Davids, M., Deforche, K., Doolabh, D., du Plessis, L., Engelbrecht, S., Everatt, J., Giandhari, J., ... de Oliveira, T. (2022). Rapid epidemic expansion of the SARS-CoV-2 Omicron variant in southern Africa. *Nature*, 603(7902), 679–686.
- Volz, E. M., Carsten, W., Grad, Y. H., Frost, S. D. W., Dennis, A. M., & Didelot, X. (2020). Identification of Hidden Population Structure in Time-Scaled Phylogenies. *Systematic Biology*, 69(5), 884–896.
- Wiley, D. C., Wilson, I. A., & Skehel, J. J. (1981). Structural identification of the antibody-binding sites of Hong Kong influenza haemagglutinin and their involvement in antigenic variation. *Nature*, 289(5796), 373–378.

Point-by-point response to reviewers' comments (manuscript 2024-01-00232B)

Referee 1

Comment 1.1: *I thank the authors for their serious attention to my comments. A simulation study definitely provides an appropriate validation of their method. In my view, the manuscript (and for that matter the response letter) still is not clear enough about what is actual validation and what is simply agreement with previous methods.*

Response to comment: We thank the reviewer for their constructive comments. We are essentially in complete agreement with their feedback and have changed the manuscript accordingly. We have answered the specific concerns below.

Comment 1.2: *The manuscript starts by saying that previous methods use "a priori" lineage definitions. This is not really a correct description since a priori means without looking at the data. It might be that the algorithms used by the approaches they discuss either are (a) not very clear or (b) involve human choice but they are still algorithms of some sort that depend critically on the data.*

Instead of "a priori" what the authors mean is that the lineages are "predefined" or better "independently defined"– i.e. defined by an another algorithm before selection coefficients are calculated for those lineages. I agree with the authors that this two-step approach may hamper the ability of the methods to identify selection signal but it is not a fundamentally different approach, it relies on the same data. In common with their method, DNA sequence data is used to identify lineages with a selective advantage.

This in clarity of definitions matters because the authors then go on to claim that agreement with previous methods validates their methods. It does not. It simply shows that they are capturing similar signals.

Response to comment: We used the term "a priori" as lineages are usually defined without data on fitness differences, e.g. in the case of *Bordetella pertussis*, only considering a few genes. Nonetheless, we agree that "a priori" could be read in a broader sense, as the reviewer points out, and will have typically involved some investigation of the data. We have therefore changed the wording as suggested.

Specific changes: This sentence now reads: "Existing methods to monitor the fitness of strains at the population level rely on independently defined strain or clade definitions, for example, Pango lineages⁴ or Nextstrain Clades⁵, the global clades for influenza⁶, or strains defined by pre-determined single mutations for *Bordetella pertussis*⁷".

Comment 1.3: *"We found that our framework was able to capture the lineage dynamics of each pathogen considered (Figure 1B-E). Using this framework on SARS-CoV-2 worldwide, we agnostically*

tracked the changes in population lineage composition, with each main Variant of Concern (VOC) having a clear change in index dynamics (Figure 1B and S2A)." To be clear, the VOCs were defined based on the same kind of data they are using, there is not a fundamental difference in approach. We do not and cannot know the truth for this real-world data. And that's a good thing for the authors. If we did know the truth, there would be no need for their methods. Indeed the manuscript seems to circle back on itself, first presenting agreement as validation then presenting differences as evidence of different power/applicability of different methods. Therefore, the simulation study should be presented first and the partial agreement with other analyses be presented as what it is, namely agreement of a formal method with previous less formal (and hence necessarily somewhat haphazard) approaches. It is entirely encouraging when a formal method can capture signals that have previously been identified by hand as it were, but (to repeat the same point again), humans are computers too.

Response to comment: We thank the reviewer for taking the time to consider the structure of the manuscript and agree that there was circularity. Ultimately, the fact that we obtained good agreement between lineages identified by our pipeline and pre-existing definitions shows that those pre-existing approaches didn't do the identification in a vacuum. We have taken the reviewer's advice and restructured this part of the paper. We now first present the simulation study. We have then reworked the language of the application to the four pathogens to present it more as an evaluation of the consistency with previous definitions (rather than those existing definitions being a way to validate our approach).

Specific changes: The first section, entitled "*Agnostic identification of lineages with underlying fitness differences*", now describes our approach and presents the simulation study. The second section is now dedicated to the "*Application to viral and bacterial pathogens*".

The text highlighted by the reviewer now reads: "We found that for each pathogen considered, *phylowave* produced evidence of lineages with clear fitness differences, as evidenced by sub-populations of genetically-related strains with discrete index dynamics (Figure 1B-E). Taking each pathogen in turn, we compared our lineage assignments with existing lineage definitions."

Comment 1.4: *As it stands, there is a very long paragraph describing the concordance of their results with previous analyses but ... long paragraphs like this are a sign of trouble and I really struggle to know what the take home lesson is. In many cases, they are comparing their selection signals with clades that were simply identified as being phylogenetic branches previously. In general concordance is not that informative about anything as I have tried to explain above, and it would be much better to focus on what are the biological conclusions from their more systematic approach. The problem of structure is also exemplified by two subheadings that the authors use that also clearly are not... actually distinct, since previous studies are amply referred to in the first subsection.*

"Agnostic identification of pathogen lineages"

"Pathogen lineages agnostically identified in the context of previous studies"

This is the main results section of the paper and sorry to say it, it's a mess.

Response to comment: We agree that the main paragraph was overly detailed. This was generated at the last round of revisions in response to another reviewer who felt additional details were needed

on each pathogen. However, on reflection, we also prefer a more concise overview of the findings, highlighting consistencies (and where there are departures). In line with the comment above, we have also reworked the subheadings in the document to reflect that we now present the simulation study and then a comparison with pre-existing lineages.

Specific changes: The first and second sections have been renamed "*Agnostic identification of lineages with underlying fitness differences*" and "*Application to viral and bacterial pathogens*", respectively. The text of the second section now reads as follows:

"We applied *phylowave* to four pathogens: SARS-CoV-2 (N=3129 global whole genome sequences), influenza H3N2 (N=1476 global hemagglutinin [HA] sequences), *B. pertussis* (N=1248 whole genome sequences from France) and *M. tuberculosis* (N=998 whole genome sequences from Samara, Russia²⁰) (Figure 1B-E and S6-S7). We found that for each pathogen considered, *phylowave* produced evidence of lineages with clear fitness differences, as evidenced by sub-populations of genetically-related strains with discrete index dynamics (Figure 1B-E). Taking each pathogen in turn, we compared our lineage assignments with existing lineage definitions.

We computed the Adjusted Rand-Index (ARI) to measure the agreement between classifications, accounting for random clustering²¹ (Figure 2 and S8). An ARI value of 1 corresponds to perfect agreement with previously defined lineages, whereas a value of 0 would be expected if clusters were assigned at random. We found that across the pathogens the level of concordance was high (ARI range 0.62-0.94). For example, the previously defined SARS-CoV-2 Variants of Concern (Alpha [B.1.1.7; 20I], Beta [B.1.351; 20H], Gamma [P.1.*; 20J], Delta [B.1.617.2/AY.*; 21A/21J], and Omicron [BA.1.1.529/BA.*; 21K]) and other previously defined sub-variants closely matched *phylowave* defined lineages (Figure 2A and S8)^{22,23}. Lineages generated by fastbaps¹⁴ and treestructure¹⁵ were less consistent with these predefined lineages (Figure S9). The existing definitions of global H3N2 clades also closely matched *phylowave* lineages (*e.g.*, 3C.3a, 3C.2a3 and 3C.2a1b.1b), with the occasional discrepancy in the exact node of emergence (*e.g.*, 3C, 3C.2 and 3C.3). *Phylowave* also identified previously defined *B. pertussis* clades (ARI = 0.63), including those defined by changes in alleles of the promoter of the pertussis toxin (*ptxP*) and fimbriae 3 gene (*fim3*)⁷. In addition, *phylowave* identified three extra *B. pertussis* lineages with clear distinct index dynamics (Figure 1D, pink, red and purple lineages), that have not been previously identified. Finally, we recovered the known lineages and sublineages (ARI = 0.92) that were present in the *M. tuberculosis* dataset^{20,24,25}.

Across the pathogens, previously defined sub-variants that reached a maximum prevalence of under 5% at any time in the datasets were generally not identified by *phylowave* (*e.g.* Eta/B.1.525, Mu/B.1.621 and EU1 for SARS-CoV-2, clades 1* of H3N2 and Central Asian Strain (CAS) and East African Indian (EAI) tuberculosis lineages). The exact limits when *phylowave* can identify discrete lineages will depend on underlying prevalence, the level of sampling and fitness differences. For example, replicating the SARS-CoV-2 analysis by continent, we did obtain *phylowave* lineages that matched previously identified variants of interest that were mainly contained to those continents and that we did not identify when using the global dataset (*e.g.* Eta/B.1.525 in Africa, Mu/B.1.621 in the Americas and EU1 in Europe) (Figure S10-11)²⁶⁻²⁸. These findings show that previous attempts to identify discrete lineages often resulted in lineage classifications with distinct fitness, even if the various algorithms to define the lineages did not use fitness as a metric."

Comment 1.5: *A second area where the manuscript could be more clear on what it is actually they are presenting. It is defined as a “framework”, which is somewhat ambiguous but it is given a name “phylowave”. Since when were frameworks given names? Phylowave seems instead to be a piece of software (or a pipeline).*

The approach would be superior from the point of scientific clarity and repeatability if it were indeed a piece of software (or pipeline) with a user manual, rather than as it is at the moment, a methods section of a paper describing an approach coupled with a set of scripts that allow the user to repeat the specific analyses they present but do not tell the user how to apply the approach to their own data and also as a corollary do not really make it obvious what choices they made in applying the framework to the data that is presented.

I don’t think in any case it is really a fully realized “framework” unless it can be applied by others. How do they envisage that this be done? Sorting this out would definitely make the manuscript a more concrete immediate contribution to the field, as well as increasing overall scientific clarity and I do not think it is justifiable to delay this until after publication.

Response to comment: We see this approach as ultimately a pipeline whereby users can take time-resolved phylogenies to identify and obtain fitness estimates for individual lineages. We decided to give it a name due to substantial interest in the approach from other groups, resulting in repeated requests to name it - we therefore prefer to keep the name *phylowave*. We agree that to facilitate the use of the pipeline there needs to be clear guidance on how to implement it. We have therefore developed step-by-step guidance using the SARS-CoV-2 dataset as an example dataset. This has now been included in the supplementary materials (Text S3). We have tested the ability of external researchers to follow these instructions, and then also apply it to their own datasets. We note that Reviewer 3 was able to replicate our analysis. In addition, in the revised manuscript, we now include general guidance on what data (including format) and parameter considerations are needed at each stage of the analysis (Text S4). These guidance documents have also been uploaded to the GitHub associated with this project.

Specific changes: The revised manuscript includes guidance on the implementation and usage of the approach (Texts S3 and S4). We have also removed references to a framework.

Comment 1.6: *A third area that should be improved is that while I appreciate the simulation study, to describe it as “big” is an exaggeration and in particular they have not tested the method until it breaks, which gives a false impression. Its more in the nature of a minimal simulation study at the moment. All the selection coefficients they have tested are very large and they should do something to ascertain what the parameter space of limited or no statistical power is, since this will be important in defining which signals can be missed by the method. This is key to interpreting the results, for real world data especially for datasets/branches when no signal is found. Also, due to detection bias, I expect the method to systematically overestimate the selection coefficients for mutations that are close to the threshold of detectability, so its important to know where that threshold is.*

Response to comment: We thank the reviewer for this comment, the threshold of detectability is indeed important. Our ability to detect a lineage with underlying fitness differences will depend on the change in fitness, the time frame considered post-emergence and the sampling intensity. In the first version of the simulation study, we focused on demonstrating the power of our method, utilising simulated datasets that were well sampled and with a sufficient timeframe for each fitness difference. We showed that *phylowave* can detect emerging lineages even in scenario with a growth rate of 0.05 per year, which is ultimately a small fitness difference - it would take 65 years for such a lineage to become the majority following its emergence. It is therefore comforting that even such small fitness differences can be identified by our approach. Nevertheless, we agree that pushing the limits of the approach until it breaks is an important part of introducing new approaches. In the revised document, we therefore expand on our simulations to consider smaller fitness and larger differences and also consider the sampling time period and the sampling intensity - as a slowly emerging pathogen may not be detectable at first, but will ultimately become detectable as greater time periods are considered. We find that we are no longer able to detect fitness differences when they are less than 0.02, even if sampled 150 years following emergence. We note that it would take 300 years for such a lineage to become the dominant lineage in a population. We further explore the impact of sampling intensity, demonstrating that the period considered is much more important than the number of samples at any point.

Specific changes: We include the following two figures (Figure S3):

*We also include the text: "To assess the performance of our approach, we repeatedly simulated phylogenetic trees where one lineage expands with a known fitness advantage compared to a background population (Figure S1). We found that *phylowave* was indeed able to identify fitter emerging lineages (Figure S2). Its ability to recover lineages depends on the time between emergence and the dates of sequences, with lineages with only small fitness advantages requiring sequences covering longer time periods (Figure S3A). Nevertheless, where the fitness difference between the two strains was greater than 0.02/year, our method was able to consistently identify the emerging lineage, noting that for lineages with lower fitness advantages the time to become the dominant lineage is >200 years. We further found that the sampling intensity was substantially less important than the sampling period (Figure S3B). Finally, we compared *phylowave* with alternative approaches (*fastbaps*¹⁴ and *treestructure*¹⁵) (Figure S4). While *treestructure* and *fastbaps* did find the emerging lineage in some cases, they always found additional lineages within the phylogenies that did not have any true selective advantage (Figure S5)."*

Figure S3: Performance of *phylowave* on datasets with varying sampling intensity and sampling window.

Ability of *phylowave* to detect an emerging lineage in datasets of different sizes **(A)** or sampling window post lineage emergence **(B)**. In **(A)** we fix the sampling window after lineage emergence to 10 years. In **(B)** we fix the size of the datasets to 1500 sequences. For each scenario, 20 trees were simulated. Details on the simulation studies can be found in Text S2.

Comment 1.7: *The authors claim to present an important new framework to address a biological problem that we can agree is high profile, important and scientifically significant. They should therefore be keen to make it as useable and interpretable as possible, this would surely increase the long-term impact of their interesting work.*

Response to comment: We agree with these concerns and, as set out in response to comment 1.5, have now included detailed guidance on the implementation and use of the pipeline. We are highly encouraged by the high level of interest we have already received on the pipeline - these initial interactions with new external users have allowed us to optimise the instructions.

Comment 1.8: *(Remarks on code availability): I did not check the code but I looked at it. It seems well organized but as I said in my review, in its present configuration it is a missed opportunity in terms of what would be useful to the community.*

Response to comment: We agree with these concerns and, as set out above, have now included detailed guidance on the implementation and use of the pipeline.

Referee 2

Comment 2.1: The authors have done an excellent job with the revisions, I have no further comments.

Response to comment: We thank the reviewer for their constructive comments which we believe made the manuscript stronger.

Referee 3

Comment 3.1: Thanks for considering my points and those of the other reviewers from the previous round of reviews, providing suitable explanations and revising the manuscript. I see that you have now named your framework `phylowave`, which I think is good and will make it much easier to cite. Also, it is good that you now include some additional simulation results from ReMaster (BEAST2) in the supplementary which I think adds to the paper. I think the revisions and additions have clarified the points well.

Response to comment: We thank the reviewer for their thorough review.

Comment 3.2: As mentioned previously, it is great that you have included the code and I am looking forward to you releasing `phylowave` as an R package.

In reviewing your existing GitHub again, I found a small thing - I needed to replace `thd.bandwidth` with `index.bandwidth` in `compute.timescale` function within `2_1_Index_computation_20231220.R` (I think you must have renamed the function for clarity for publication).

Response to comment: Thank you, this has been updated in the GitHub repository.